# Context-guided Embedding Adaptation for Effective Topic Modeling in Low-Resource Regimes

**Yishi Xu**,* **Jianqiao Sun**,* **Yudi Su, Xinyang Liu, Zhibin Duan, Bo Chen**,†
National Key Laboratory of Radar Signal Processing, Xidian University, Xi'an, China, 710071
{xuyishi, jianqiaosun}@stu.xidian.edu.cn, bchen@mail.xidian.edu.cn

**Mingyuan Zhou**
McCombs School of Business, The University of Texas at Austin, TX 78712, USA
mingyuan.zhou@mccombs.utexas.edu

## Abstract

Embedding-based neural topic models have turned out to be a superior option for low-resourced topic modeling. However, current approaches consider static word embeddings learnt from source tasks as *general knowledge* that can be transferred directly to the target task, discounting the dynamically changing nature of word meanings in different contexts, thus typically leading to sub-optimal results when adapting to new tasks with unfamiliar contexts. To settle this issue, we provide an effective method that centers on adaptively generating semantically tailored word embeddings for each task by fully exploiting contextual information. Specifically, we first condense the contextual syntactic dependencies of words into a semantic graph for each task, which is then modeled by a *Variational Graph Auto-Encoder* to produce task-specific word representations. On this basis, we further impose a learnable Gaussian mixture prior on the latent space of words to efficiently learn topic representations from a clustering perspective, which contributes to diverse topic discovery and fast adaptation to novel tasks. We have conducted a wealth of quantitative and qualitative experiments, and the results show that our approach comprehensively outperforms established topic models.

## 1 Introduction

The last two decades have witnessed the enduring popularity of topic models along with their many successful applications in a range of fields [1–7]. And this is predominantly attributed to their ability to reveal the underlying semantic structure from large volumes of textual data. By identifying a group of salient themes, topic models represent each document as a mixture of them, providing an intuitive understanding of the target corpus. Although conventional probabilistic topic models [8–14] have been widely used, new variants continue to spring up in the era dominated by deep neural networks.

Among the proliferation of new methodologies come both the neural topic models (NTMs) [15–18] resulting from the development of variational autoencoders (VAEs) and autoencoding variational Inference (AVI) [19, 20], and the contextualized topic models (CTMs) [21–24] benefiting from the flourishing of pre-trained language models [25, 26]. However, these recently developed approaches essentially maintain the assumption of sufficient resources, *i.e.*, with a plethora of documents being available. Comparatively, little attention has been paid to topic modeling under resource-limited or resource-poor conditions [27], which plays a significant role in real-world applications.

---

*Equal contribution
†Corresponding author

37th Conference on Neural Information Processing Systems (NeurIPS 2023).

A case in point occurs in personalized recommendation systems where users' preferences are judged based on only a small amount of their historical data, such as past purchases or online behaviors [28]. Another example arises from crisis management. Since being able to quickly identify and monitor emerging topics during a crisis, *e.g.*, a public health emergency, could substantially support a government or organization in responding appropriately to rapidly changing situations [29].

While there have also been some beneficial attempts to study few-shot learning for topic modeling, they more or less exhibit certain limitations. For instance, Iwata [27] first proposed a straightforward method that aims to learn good, *i.e.*, task-specific priors for latent Dirichlet allocation (LDA) [8] by using neural networks. However, this approach suffers from a lack of expressiveness when dealing with difficult novel tasks. Later on, Duan et al. [30] claimed that embedding-based NTMs [31] are naturally superior in generalizing to new and unseen tasks. By considering word embeddings learned from the training task pool as transferable knowledge, they have shown that effective generalization can be achieved by only learning topic embeddings adaptively. Nevertheless, as the word semantics inevitably change with contexts, the learned static word embeddings may not adapt well to the target task with alien contexts. On the other hand, we have experimentally found that the performance of two representative CTMs [21, 22] is also not competitive enough under resource-poor conditions.

To address the above issues, we propose to learn adaptive word embeddings suitable for each task by fully exploiting the contextual grammar information. Concretely, we first construct a task-specific semantic graph between words using well-established dependency parsing tools[3] [32]. This graph helps depict the precise meaning of each word in the given task, which we then model with a created variational graph autoencoder (VGAE) [33], and the resulting latent representations of words merge the contextual information and are thus semantically tailored to the given task. Furthermore, by imposing a Gaussian mixture prior on the latent space of word, we offer a perspective on learning topics through clustering, *i.e.*, each component of the Gaussian mixture can be viewed as the representation of a topic. Consequently, the adaptive word embeddings tend to be reasonably covered by several clusters, facilitating the discovery of more interpretable topics closely related to the given task.

Our main contributions can be summarized as follows:

- To solve the problem of few-shot learning for topic modeling, we propose to learn adaptive word embeddings that are semantically matched to the given task.

- To generate adaptive word embeddings for each task, we innovatively introduce a variational graph autoencoder to learn the latent representations of words from the task-specific semantic graph that captures the contextual syntactic dependencies.

- To mine more interpretable topics related to the given task, we offer a perspective on learning topics through clustering by imposing a Gaussian mixture prior on the word latent space. Besides, we also develop an efficient variational inference scheme to approximate the posteriors of latent variables.

## 2 Meta-CETM

### 2.1 Problem Formulation

To clarify the problem definition of few-shot learning for topic modeling, we assume that there are $C$ training corpora $\{\mathcal{D}_c\}_{c=1}^C$ from different domains, with the goal of using them to learn a topic model that generalizes to the test corpus $\mathcal{D}_{test}$ in a new domain, *i.e.*, the resulting topic model is supposed to quickly adapt to a given test task with a few documents and mine topics related to the new domain. Further, we adopt an episodic training strategy as is conventional in most few-shot learning literature. Specifically, we construct a batch of training tasks $\{\mathcal{T}^{(i)}\}_{i=1}^M$ to mimic the scenario of giving tasks at test time, so that each task contains only several documents from an arbitrary training corpus. And we denote the bag-of-words (BoW) representations of the task documents as $\mathbf{X}^{(i)} \in \mathbb{R}^{V \times J}$, where $V$ is the vocabulary size and $J$ is the number of documents in each task. In addition, our method also builds a task-specific dependency graph whose adjacency matrix is represented as $\mathbf{A}^{(i)} \in \mathbb{R}^{V \times V}$. To understand the concepts of corpus, task, and document more clearly, see the example in Appendix C.

---

[3]The semantic graphs are built with the help of spaCy, a public natural language processing tool to analyse the grammatical structure of sentences, `https://spacy.io/`.

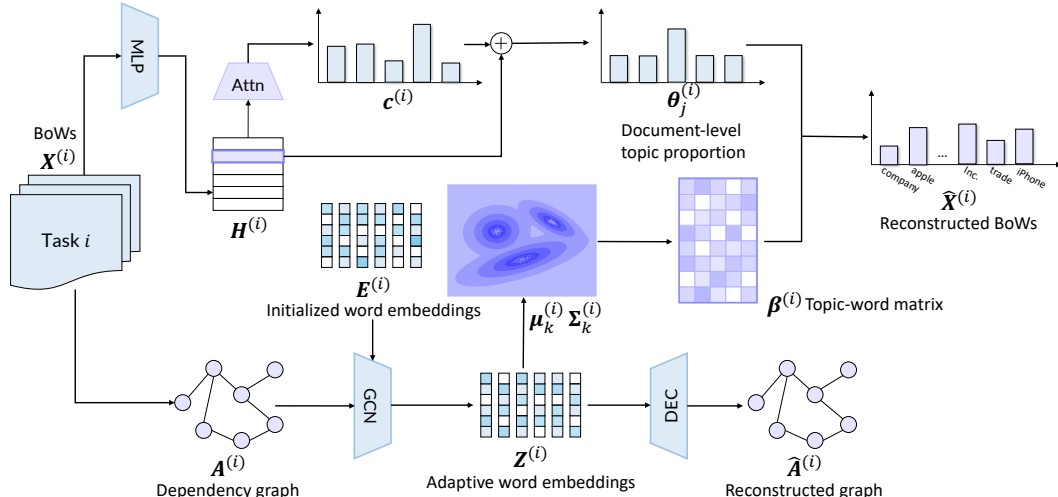

Figure 1: Overview of the proposed method. The top branch establishes a standard neural topic modeling pipeline, and the bottom branch creates a graph VAE to learn contextualized word embeddings, with a Gaussian mixture prior imposed on the latent space to yield task-specific topic representations. Note that the topic-word matrix is derived based on the probability density assigned to the adaptive word embeddings.

## 2.2 Generative Model

In this section, we present a **C**ontextualized **E**mbedded **T**opic **M**odel, dubbed as **Meta-CETM**, to cope with the problem of topic modeling under resource-poor conditions, the essence of which is to make extra use of the contextual information of given documents to learn dynamic word embeddings that are well adapted to the current task. Concretely, for any given task $\mathcal{T}^{(i)}$, in addition to obtaining the BoW representations $\mathbf{X}^{(i)}$ of its documents, which only imply the word co-occurrence patterns, we also build a semantic graph between words based on the contextual dependency grammars. Thus the corresponding adjacency matrix $\mathbf{A}^{(i)}$ can be viewed as complementary information on the word semantics. With $\mathbf{X}^{(i)}$ and $\mathbf{A}^{(i)}$, then the goal of the generative model is to model these two types of observations jointly, whose key lies in how to establish a bridge connecting the two.

Inspired by the recently developed VGAEs [33] and embedded topic models (ETMs) [18, 31], we create our generative model based on the compact assumption, *i.e.*, there exists a shared latent space of words that can be mapped to the observation spaces of BoW and semantic graph, respectively. Following this conception, we portray a well-structured generative process via a specific task $\mathcal{T}^{(i)}$. First, the latent representations of words $\mathbf{Z}^{(i)} \in \mathbb{R}^{D \times V}$ are sampled from a prior distribution. Acting as the suitable connection, they are responsible for generating both the adjacency matrix $\mathbf{A}^{(i)}$ and the BoW $\mathbf{X}^{(i)}$. To produce $\mathbf{A}^{(i)}$, we use a typical inner product graph decoder formulated as

$$\mathbf{A}^{(i)} \sim \mathrm{Ber}(\sigma(\mathbf{Z}^{(i)^\top}\mathbf{Z}^{(i)})) \tag{1}$$

As for the generation of BoW $\mathbf{X}^{(i)}$, the latent representations $\mathbf{Z}^{(i)}$ (or word embeddings) are mainly used to derive the topic-word matrix. Unlike previous ETMs that usually decompose the topic-word matrix into the inner product of learnable word embeddings and topic embeddings, here we provide a Bayesian perspective on interpreting the topic-word matrix by imposing a Gaussian mixture prior to the word latent space. More precisely, we regard each component of the Gaussian mixture prior as the representation of a topic, such that the word embeddings are naturally generated from different clusters (topics). And the prior probability for the $v^{th}$ word embedding is given by

$$p(\mathbf{z}_v^{(i)}) = \sum_{k=1}^{K} \pi_k \cdot \mathcal{N}(\mathbf{z}_v^{(i)}|\boldsymbol{\mu}_k, \boldsymbol{\Sigma}_k), \tag{2}$$

where $\pi_k$ is the prior mixing coefficient of the $k^{th}$ topic and $\sum_k \pi_k = 1$. As such, each topic can be represented as a Gaussian distribution in the word latent space, with a probability density assigned to each word embedding. This provides a natural choice to define the topic-word matrix $\boldsymbol{\beta}^{(i)} \in \mathbb{R}^{V \times K}$.

Hence, the distribution of topic $k$ over the vocabulary $\boldsymbol{\beta}_k^{(i)} \in \mathbb{R}^V$ can be derived by

$$\boldsymbol{\beta}_k^{(i)} = \text{Softmax}(p(\mathbf{Z}^{(i)}|\boldsymbol{\mu}_k, \boldsymbol{\Sigma}_k)), \tag{3}$$

In this way, we expect that semantically similar words will be close in the latent space and generated from the same topic. Moreover, we also posit a context variable $\boldsymbol{c}^{(i)}$ that summarizes the task-level information about the topic proportion and thus serves as a prior for generating the topic proportion $\boldsymbol{\theta}_j^{(i)}$ of each document. Formally, the complete generation process of our model can be formulated as

1. Generate the adjacency matrix of semantic graph using Eq. 1;

2. For the $v^{th}$ word appeared in the task $\mathcal{T}^{(i)}$:

   (a) Draw its topic assignment $y_v^{(i)} \sim \text{Cat}(\boldsymbol{\pi})$;

   (b) Draw its word embedding $\mathbf{z}_v^{(i)} \sim \mathcal{N}(\boldsymbol{\mu}_{y_v^{(i)}}, \boldsymbol{\Sigma}_{y_v^{(i)}})$;

3. Compute task-specific topic-word matrix $\boldsymbol{\beta}^{(i)}$ based on Eq. 3;

4. Draw task-level context variable $\boldsymbol{c}^{(i)} \sim \mathcal{N}(0, a^2\boldsymbol{I})$;

5. For the $j^{th}$ document in the task $\mathcal{T}^{(i)}$:

   (a) Draw its topic proportion $\boldsymbol{\theta}_j^{(i)} \sim \mathcal{LN}(\boldsymbol{c}^{(i)}, b^2\boldsymbol{I})$;

   (b) For the $n^{th}$ word in the $j^{th}$ document:

      i. Draw its topic assignment $e_{jn}^{(i)} \sim \text{Cat}(\boldsymbol{\theta}_j^{(i)})$;

      ii. Generate the word count $x_{jn}^{(i)} \sim \text{Cat}(\boldsymbol{\beta}_{e_{jn}^{(i)}}^{(i)})$,

where $\text{Cat}()$, $\mathcal{N}()$, $\text{Ber}()$, and $\mathcal{LN}()$ denote categorical, Gaussian, Bernoulli, and logistic-normal distributions, respectively. $\sigma$ is the $\text{Sigmoid}()$ operation[4]. $a$ and $b$ are both hyper-parameters. Table 4 in Appendix B gives a list of key notations used in this paper.

## 2.3 Variational Inference Algorithm

Observing a task $\mathcal{T}^{(i)}$ with data $\mathbf{X}^{(i)}$, the goal of inference network is to approximate the posterior distributions over the latent variables, $\{\boldsymbol{\theta}_j^{(i)}, \boldsymbol{c}^{(i)}, \mathbf{Z}^{(i)}\}$, and the parameters of GMM, $\{\pi_k^{(i)}, \boldsymbol{\mu}_k^{(i)}, \boldsymbol{\Sigma}_k^{(i)}\}$.

**Document-specific latent variable inference.** To model the uncertainty of topic proportions, we define the variational posterior distribution for $\boldsymbol{\theta}_j^{(i)}$, and a residual multi-layer perception (MLP) is employed to learn distribution parameters. To be specific,

$$q(\boldsymbol{\theta}_j^{(i)}|\boldsymbol{x}_j^{(i)}, \boldsymbol{c}^{(i)}) = \mathcal{LN}(\boldsymbol{\mu}_{\boldsymbol{\theta}_j^{(i)}}, \boldsymbol{\Sigma}_{\boldsymbol{\theta}_j^{(i)}})$$
$$\boldsymbol{\mu}_{\boldsymbol{\theta}_j^{(i)}}, \boldsymbol{\Sigma}_{\boldsymbol{\theta}_j^{(i)}} = \text{ResMLP}(\mathbf{h}_j^{(i)} + \boldsymbol{c}^{(i)}) \tag{4}$$
$$\mathbf{h}_j^{(i)} = \text{MLP}_{\mathbf{h}}(\mathbf{x}_j^{(i)})$$

where $\boldsymbol{\mu}_{\boldsymbol{\theta}_j^{(i)}}$ and $\boldsymbol{\Sigma}_{\boldsymbol{\theta}_j^{(i)}}$ are logistic Gaussian distribution deterministic parameters depending on the latent mean $\boldsymbol{c}^{(i)}$ and the document latent representation $\mathbf{h}_j^{(i)}$ of $\mathbf{x}_j^{(i)}$.

**Task-specific latent variable inference.** For topic proportion mean $\boldsymbol{c}^{(i)}$, we apply Gaussian distribution to approximate the variational posterior, whose parameters mean vectors and covariance matrices are derived through encoding all documents in task $\mathcal{T}^{(i)}$ with attention mechanism [34], written as

$$q(\boldsymbol{c}^{(i)}|\mathbf{X}^{(i)}) = \mathcal{N}(\boldsymbol{\mu}_{\boldsymbol{c}^{(i)}}, \boldsymbol{\Sigma}_{\boldsymbol{c}^{(i)}}); \boldsymbol{\mu}_{\boldsymbol{c}^{(i)}}, \boldsymbol{\Sigma}_{\boldsymbol{c}^{(i)}} = \text{Attn}(\mathbf{H}^{(i)}); \mathbf{H}^{(i)} = \text{MLP}(\mathbf{X}^{(i)}), \tag{5}$$

where $\mathbf{H}^{(i)}$ is the latent indication of the task, and $\text{Attn}()$ is the attention mechanism to capture relationships of $J$ documents in task $\mathcal{T}^{(i)}$.

---

[4]If a weighted graph $\mathbf{A}^{(i)}$ is considered, then the Poisson likelihood can be employed with $\sigma$ set to the $\text{Softplus}()$ operation.

Additionally, to ensure the modeling flexibility, we design a simple inference network consisting of a two-layer GCN [35] to infer the latent representations $\mathbf{Z}^{(i)}$ of words following [33]:

$$q(\mathbf{Z}^{(i)}|\mathbf{A}^{(i)}, \mathbf{E}^{(i)}) = \mathcal{N}(\mathbf{Z}^{(i)}|\boldsymbol{\mu}_{\mathbf{Z}}^{(i)}, \boldsymbol{\Sigma}_{\mathbf{Z}}^{(i)})$$
$$\boldsymbol{\mu}_{\mathbf{Z}}^{(i)}, \boldsymbol{\Sigma}_{\mathbf{Z}}^{(i)} = \text{GCN}(\mathbf{A}^{(i)}, \mathbf{E}^{(i)}) \tag{6}$$

where $\mathbf{E}^{(i)}$ is the initialized word features and $\mathbf{A}^{(i)}$ is derived through a neural parser on $\mathbf{X}^{(i)}$. Figure 1 illustrates an overview of our approach, including the variational inference network.

In this paper, for simplicity, we denote $\Psi$ as network parameters of both the encoder and the decoder.

**Expectation Maximization for solving $\boldsymbol{\mu}_k^{(i)}$ and $\boldsymbol{\Sigma}_k^{(i)}$.** As discussed before, $\boldsymbol{\mu}_k^{(i)}$ and $\boldsymbol{\Sigma}_k^{(i)}$ are task-specific parameters of Gaussian mixture distribution, which are not learnable variables, and we do not have analytic solutions for Maximum Likelihood Estimation (MLE) of GMM containing the non-differentiable sampling process [36]. To approximate the posterior, we resort to Expectation Maximization (EM) [37] algorithm to optimize the parameters, formulated as:

$$\text{E} - \text{Step} \quad Q_v^{(i)} := p(y_v^{(i)} = k|\mathbf{z}_v^{(i)}) = \frac{\pi_k^{(i)}\mathcal{N}(\mathbf{z}_v^{(i)}; \boldsymbol{\mu}_k^{(i)}, \boldsymbol{\Sigma}_k^{(i)})}{\sum_k \pi_k^{(i)}\mathcal{N}(\mathbf{z}_v^{(i)}; \boldsymbol{\mu}_k^{(i)}, \boldsymbol{\Sigma}_k^{(i)})}$$

$$\text{M} - \text{Step} \quad \boldsymbol{\mu}_k^{(i)} := \frac{\sum_v Q_v^{(i)}\mathbf{z}_v^{(i)}}{\sum_v Q_v^{(i)}}$$

$$\boldsymbol{\Sigma}_k^{(i)} := \frac{\sum_v Q_v^{(i)}(\mathbf{z}_v^{(i)} - \boldsymbol{\mu}_k^{(i)})(\mathbf{z}_v^{(i)} - \boldsymbol{\mu}_k^{(i)})^T}{\sum_v Q_v^{(i)}} \tag{7}$$

$$\pi_k^{(i)} := \frac{\sum_v Q_v^{(i)}}{\sum_k \sum_v Q_v^{(i)}}.$$

Since topic $k$ is sampled from the Uniform distribution for each task, the mixing coefficients are initialized as $\frac{1}{K}$. We initialize $\boldsymbol{\mu}_k^{(i)}$ and $\boldsymbol{\Sigma}_k^{(i)}$ as the average of latent variables $\mathbf{Z}^{(i)}$ and the identity matrix $\boldsymbol{I}$, respectively. Here, we only display the final updating formulas. The detailed derivation processes for E-step and M-step are presented in Appendix E.2.

## 2.4  Training Objective and Optimization

By Jensen's inequality, the evidence lower bound (ELBO) of each task can be derived as

$$\mathcal{L}_{ELBO} = \sum_{j=1}^{J} \mathbb{E}_Q \left[ \log p(\mathbf{x}_j^{(i)} \mid \boldsymbol{\theta}_j^{(i)}, \mathbf{Z}^{(i)}) \right] + \sum_{j=1}^{J} \mathbb{E}_Q \left[ \log \frac{p(\boldsymbol{\theta}_j^{(i)} \mid \boldsymbol{c}^{(i)})}{q(\boldsymbol{\theta}_j^{(i)} \mid \mathbf{x}_j^{(i)}, \boldsymbol{c}^{(i)})} \right]$$
$$+ \mathbb{E}_Q \left[ \log \frac{p(\boldsymbol{c}^{(i)})}{q(\boldsymbol{c}^{(i)} \mid \mathbf{X}^{(i)})} \right] + \mathbb{E}_Q \left[ \log p(\mathbf{A}^{(i)} \mid \boldsymbol{Z}^{(i)}) \right] + \mathbb{E}_Q \left[ \log \frac{p(\mathbf{Z}^{(i)})}{q(\mathbf{Z}^{(i)} \mid \mathbf{A}^{(i)}, \mathbf{E}^{(i)})} \right] \tag{8}$$

where

$$Q = \prod_{j=1}^{J} q(\boldsymbol{\theta}_j^{(i)} \mid \mathbf{X}^{(i)}, \boldsymbol{c}^{(i)}) q(\boldsymbol{c}^{(i)} \mid \mathbf{X}^{(i)}) q(\mathbf{Z}^{(i)} \mid \mathbf{A}^{(i)}, \mathbf{E}^{(i)}) \tag{9}$$

is the variational joint distribution. The first and the fourth terms in Eq. 8 are the reconstruction errors for document BoW and the graph adjacency matrix, respectively. The remaining three terms are all Kullback–Leibler (KL) divergence to constrain the distance between the prior distribution and the variational posterior distribution. Owing to the space limit, we only present the final formulas for $\mathcal{L}_{ELBO}$ here. The detailed derivations, the training algorithm, and the meta-testing algorithm can be found in Appendix E.1, Alg. 1, and Alg. 2, respectively.

## 3  Experiments and Analysis

### 3.1  Experimental setup

**Datasets.** We conducted experiments on four widely used textual benchmark datasets, specifically *20Newsgroups* (**20NG**) [38], *Yahoo Answers Topics* (**Yahoo**) [39], *DBpedia* (**DB14**) [40], and *Web of Science* (**WOS**) [41].

Table 1: PPL results on four datasets. "5" and "10" denote the number of documents in each task. *Since ProdLDA and ETM are not designed for few-shot learning, we run their meta versions where parameters are optimized using model-agnostic meta-learning (MAML) strategy [44].

| Methods | 20NG | | Yahoo | | DB14 | | WOS | |
|---|---|---|---|---|---|---|---|---|
| | 5 | 10 | 5 | 10 | 5 | 10 | 5 | 10 |
| LDA [42] | 4021±1528 | 3502±1277 | 4476±1544 | 4028±1097 | 4410±1918 | 3697±1747 | 3439±671 | 3246±461 |
| PFA [12] | 3463±1452 | 3150±1119 | 3257±1328 | 3122±1040 | 3443±1937 | 3170±1562 | 3113±819 | 3431±830 |
| ProdLDA [43] | 4853±1034 | 4523±817 | 5765±1104 | 5378±826 | 5477±846 | 5297±740 | 4311±469 | 4220±392 |
| ETM [18] | 3192±895 | 3107±671 | 2868±909 | 2817±620 | 3217±1960 | 3054±1539 | 3135±704 | 3310±455 |
| MAML-ProdLDA* | 4292±1123 | 4355±997 | 4354±1369 | 4250±919 | 4844±1337 | 4678±1119 | 4117±462 | 4068±332 |
| MAML-ETM* | 3849±1064 | 3725±841 | 3653±1081 | 3642±776 | 4448±2737 | 4279±2301 | 3483±4044 | 3277±644 |
| Meta-SawETM [30] | 2872±869 | 2984±740 | 2365±934 | 2487±756 | 2047±1374 | 1914±1009 | 2031±445 | 2253±315 |
| CombinedTM [21] | 2660±659 | 2595±625 | 2700±590 | 2674±575 | 1851±767 | 1774±731 | 2562±633 | 2648±658 |
| ZeroShotTM [22] | 2904±851 | 2569±663 | 2822±732 | 2795±721 | 1938±758 | 1835±739 | 2863±704 | 2775±558 |
| **Meta-CETM** | **954**±543 | **1170**±606 | **1074**±442 | **1219**±455 | **802**±571 | **1084**±643 | **1293**±542 | **1528**±218 |

**Baseline methods.** Our model is compared with exemplary baseline methods, including probabilistic topic models, state-of-the-art NTMs and CTMs. Specifically, we conduct experiments of the following methodologies under the document-limited setting: 1) **LDA** [42]; 2) **PFA** [12]; 3) LDA with Products of Experts (**ProdLDA**) [43], which replaces the mixture model in LDA with products of experts and updates parameters using AVI; 4) Embedded Topic Model (**ETM**) [18], an NTM incorporating word embeddings and learning with AVI. For a fair comparison, we also consider the variations of ProdLDA and ETM under the few-shot setting, referred to 5) **MAML-ProdLDA** and 6) **MAML-ETM**, where the parameters are optimized through MAML algorithm [44]. In addition, we include a model-based hierarchical NTM, 7) **Meta-SawETM** [30], but we only apply their single-layer model. Moreover, another two CTMs 8) **CombinedTM** [21] and 9) **ZeroShotTM** [22] are also compared, both of which contain contextualized Sentence BERT [45] embeddings as the model input.

## 3.2 Experimental results and analysis

In this section, we evaluate the predictive performance, topic quality, and classification performance of our model through an extensive series of experiments. Note that in all tables, we have highlighted the best and runner-up results in boldface and with an underline, respectively. Our code is available at `https://github.com/NoviceStone/Meta-CETM`.

### 3.2.1 Per-holdout-word perplexity

Following the practice in Meta-SawETM [30], we adopt the per-holdout-word perplexity (PPL) [46] to measure the predictive performance of our model. Specifically, for each task composed of several documents, $80\%$ of the tokens in the BoWs are randomly chosen to form a support set $D_{test}^S$, which is used to adapt to a task-specific topic-word matrix $\beta$, and the remaining $20\%$ word tokens are held out to form the query set $D_{test}^Q$ with data $\mathbf{Y}$. Then, the PPL can be calculated as

$$\exp\left\{-\frac{1}{y_{..}}\sum_{v=1}^{V}\sum_{n=1}^{N} y_{vn} \ln \frac{\sum_{s=1}^{S}\sum_{k=1}^{K} \beta_{vk}^s \theta_{kn}^s}{\sum_{s=1}^{S}\sum_{v=1}^{V}\sum_{k=1}^{K} \beta_{vk}^s \theta_{kn}^s}\right\}, \tag{10}$$

where $S$ is the total number of collected samples and $y_{..} = \sum_{v=1}^{V}\sum_{n=1}^{N} y_{vn}$.

**Model settings.** For all compared methods, we set the number of topics as 10. And for all NTMs, the hidden layers size of the encoder is set to 300. For all embedding-based topic models, *i.e.,* ETM, MAML-ETM, Meta-SawETM and our Meta-CETM, we load pretrained GloVe word embeddings [47] as the initialization for a fair comparison. Finally, We train our model using the Adam optimizer [48] with a learning rate of $1 \times 10^{-2}$ for 10 epochs on an NVIDIA GeForce RTX 3090 graphics card.

**Results.** In Table 1, we list the PPL of ten compared methods on four datasets. It can be noticed although LDA and PFA are both traditional probabilistic topic models, PFA presents better results than LDA. By utilizing MAML to learn parameter initializations, MAML-ProdLDA performs better than ProdLDA while MAML-ETM exhibits poorer results than ETM. This can be attributed to that ETM possesses more parameters than ProdLDA, and calculating the gradients in a high-dimensional space with only a few documents is difficult for MAML-ETM. Applying Weibull distribution to model the latent representation for documents and employing task-specific variable designs for both

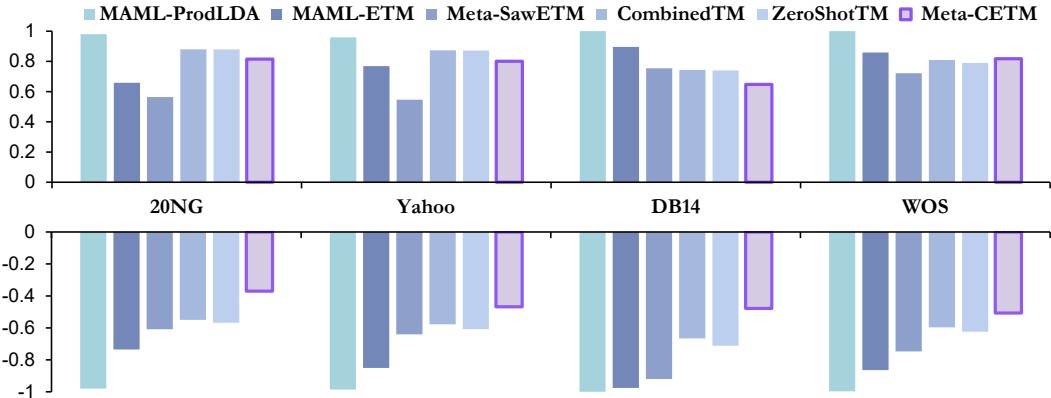

Figure 2: Topic diversity results (top row) and topic coherence results (bottom row) on four datasets of six compared methods. The number of documents for each task is 10.

topic proportions and topic matrices, Meta-SawETM acts superior to other topic models. Moreover, incorporating contextualized embeddings as input enables CombinedTM and ZeroShotTM to present competitive performance. Furthermore, it is noteworthy that Meta-CETM achieves the lowest PPL among all these methods, indicating the excellent predictive performance of our model.

### 3.2.2 Topic quality

In this part, we evaluate the topic quality of different methods in terms of the topic diversity (**TD**) [49], defined as the percentage of unique words in the top 10 words of all topics, and the topic coherence (**TC**), which measures the average Normalized Pointwise Mutual Information (NPMI) [50, 51] values to count word co-occurrences[5]. The experimental settings are the same as in Sec. 3.2.1. For each task, the number of documents is 10 for all datasets. The results are displayed in Fig. 2 and it can be notably and interestingly observed that MAML-ProdLDA achieves fairly "perfect" TD results, but it shows the worst TC performances among compared methods. Such inconsistency is brought by MAML-ProdLDA's concentration on a large amount of universal and frequently-occurring words when training on the base data, hindering it from extracting informative topics given the meta-test task. Embodying the embedding design for words and topics, MAML-ETM and Meta-SawETM are more likely to mine context-related topics than MAML-ProdLDA. Besides, as the representatives of CTM, CombinedTM and ZeroShotTM equipped with BERT embeddings present more comprehensive performances than previous topic models. From the TC perspective, our Meta-CETM yields the most favorable results among the six methods, indicating the fast adaptability in discovering interpretable topics with limited documents.

### 3.2.3 Few-shot document classification

To further validate that our model is capable of learning topics which are highly adapted to the task, we undertake experiments on few-shot document classification.

**Model settings.** As in Meta-SawETM [30], we compare our Meta-CETM with classical meta-learning algorithms under different architectures. Specifically, we design a three-layer feedforward network as **MLP** structure and three-layer 1-dimensional convolutions followed by batch normalization, $\mathrm{ReLU}()$ activation and max-pooling operation as **CNN**. For meta-learning methods, we apply **MAML** [44] to learn parameter initializations, and prototypical network (**PROTO**) [52] to learn an embedding space and minimize the distance between the clustering centroids and the samples. Besides, we adopt two fine-tuning manners, named **FT** [53] and **FT***, to update the classifier parameters and all parameters of the model, respectively. Single-layer HNS-SawETM and single-layer Meta-SawETM in [30] are compared as well. Additionally, we investigate the CombinedTM [21] and ZeroShotTM [22] to evaluate their few-shot classification performance. Different from PPL evaluation in Sec. 3.2.1, the support set and the query set for classification are sourced from two batches of documents.

---

[5]The NPMI score is computed through the *gensim* package in `https://radimrehurek.com/gensim/models/coherencemodel.html`.

Table 2: 5-way 5-shot and 5-way 10-shot few-shot document classification results on all four datasets. *denotes all parameters of the model are fine-tuned.

| Methods | | 20NG | | DB14 | | Yahoo | | WOS | |
|---|---|---|---|---|---|---|---|---|---|
| Rep. | Alg. | 5 shot | 10 shot | 5 shot | 10 shot | 5 shot | 10 shot | 5 shot | 10 shot |
| MLP | MAML [44] | 32.01 | 36.20 | 50.20 | 60.30 | 45.42 | 51.00 | 37.77 | 40.43 |
| | PROTO [52] | 35.20 | 38.30 | 54.13 | 57.16 | 50.01 | 56.16 | 39.61 | 41.46 |
| | FT [53] | 29.70 | 33.04 | 51.11 | 53.83 | 48.59 | 53.06 | 36.52 | 37.22 |
| | FT* | 38.87 | 48.52 | 71.12 | 77.94 | 50.73 | 56.74 | 45.02 | 51.20 |
| CNN | MAML [44] | 34.08 | 45.40 | 66.28 | 75.96 | 48.81 | 56.50 | 47.28 | 57.32 |
| | PROTO [52] | 39.86 | 49.71 | **78.58** | **81.01** | 53.16 | 63.66 | 59.05 | **67.75** |
| | FT [53] | 45.70 | 53.63 | 74.68 | 80.75 | 56.78 | 66.04 | 54.68 | 63.39 |
| | FT* | 44.53 | 51.92 | 72.49 | 80.07 | 53.28 | 52.56 | 51.42 | 61.98 |
| HNS-SawETM [30] | | 39.37 | 43.78 | 65.93 | 71.08 | 52.35 | 57.86 | 42.09 | 56.91 |
| Meta-SawETM [30] | | 39.19 | 45.83 | 67.20 | 72.31 | 52.45 | 60.58 | 43.39 | 57.44 |
| CombinedTM [21] | | 46.17 | 52.73 | 68.42 | 73.26 | 57.94 | 64.75 | 56.16 | 65.97 |
| ZeroShotTM [22] | | 46.65 | 52.08 | 71.93 | 76.09 | 58.12 | 66.21 | 58.50 | 66.10 |
| Meta-CETM | | **50.57** | **58.47** | 76.85 | 79.34 | **63.84** | **72.67** | **61.47** | 67.62 |

**Classification strategy.** For typical few-shot learning algorithms, we follow the convention to train parameters of the feature extractor in a supervised manner. At meta-test stage, we use the support set (*i.e.,* a few labeled examples) to adapt to a task-specific classifier and compute the accuracy on the query set. As for topic models, where no dedicated classifiers are available, we first use the support set to learn a group of class-specific topic-word matrices, then for each document in the query set, we calculate its reconstruction error as the basis for classification.

**Results.** The classification results[6] are listed in the Table 2. As we can see, the CNN architecture outperforms MLP under the same algorithms, which can be attributed to CNN's unique inductive bias of the locality. Furthermore, PROTO surpasses MAML by a large margin in most cases, indicating that a good embedding space is more useful for classification than favorable parameter initializations. For the CNN architecture, fine-tuning the classifier only (FT) and fine-tuning all parameters (FT*) make slight differences; but for MLP, we observe a considerable performance boost by updating all parameters of the network over the FT algorithm. We postulate that the classification results of MLP are more susceptible to the variation of feature extractor parameters due to its linear structure. Additionally, CombinedTM and ZeroShotTM, incorporating contextualized representations, outperforms HNS-SawETM and Meta-SawETM by learning class-specific topic-word matrices more effectively. With the design of adaptive word embeddings, our Meta-CETM not only achieves much better results than previous topic models, but also is comparable to few-shot learning algorithms particularly designed for supervised learning.

### 3.2.4 Embedding space visualization

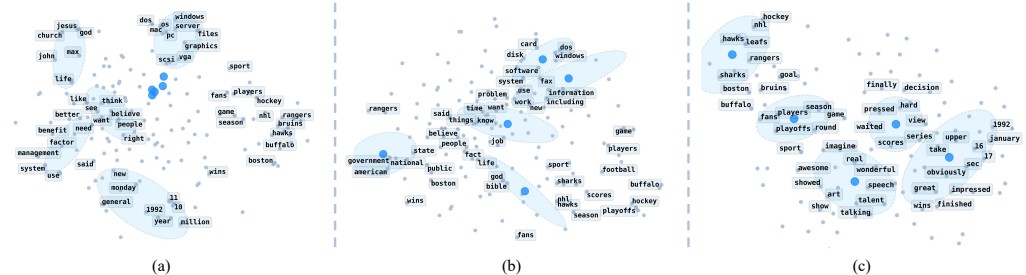

Figure 3: Visualization of the adapted embedding space for (a) MAML-ETM, (b) Meta-SawETM and (c) Meta-CETM. The small grey points represent word embeddings, and the big blue points denote topic embeddings for MAML-ETM, topic means for Meta-SawETM and our Meta-CETM. For Meta-SawETM and our Meta-CETM, the ellipse coverages represent topic covariances. For MAML-ETM, the ellipse coverages are the areas of top words. The target task is sampled from the sub-topic "rec.sport.hockey" of 20NG dataset.

---

[6] The classification experiments for all compared methods are conducted ten times.

In addition to quantitative results, we also qualitatively analyzed the effectiveness of our model by visualizing the adapted embedding space, as shown in Figure 3. It can be seen that both MAML-ETM and Meta-SawETM learn topics that are not highly relevant to the target task, as most of their top words (*e.g., windows* and *dos*) are inherited from the base training corpora, while those informative words (*e.g., players* and *hockey*) associated with the target task are away from the topic embeddings. By contrast, in the embedding space learned by Meta-CETM, the adapted Gaussian distributions, *i.e.,* topics reasonably cover almost all words closely related to the target task, indicating that our model can achieve successful adaptation and discover more interpretable topics.

Furthermore, we investigated whether the adaptive word embeddings generated by our model effectively reflect the contextual information of each given task. As illustrated in Figure 4, the word embedding of "apple" adapted from a task related to *company* is surrounded by words like "mobile" and "amazon", whereas in another task concerning *plant*, the embedding of "apple" is closer to the words such as "fruit" and "trees". This phenomenon suggests that Meta-CETM produces word embeddings that are semantically well-matched to the context of the target task.

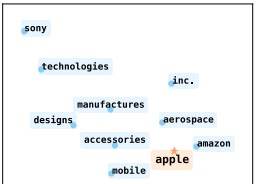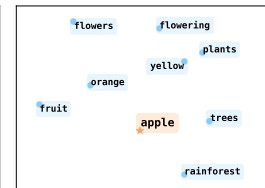

Figure 4: The adaptive contextual word embeddings learned by our Meta-CETM on DB14 dataset [40]. Left: The local embedding space of a task from the "Company" domain, Right: The local embedding space of a task in the "Plant" domain.

### 3.3 Ablation study

To explore the effectiveness of key designs in our model, we conduct a series of ablation experiments on all four datasets by removing each designed module separately. The numerical results of perplexity, topic diversity, and topic coherence are listed in Table 3, where ETM [18] is chosen as the baseline method. From the results presented below, we have the following observations:

*i)* Despite achieving the best TD results, ETM obtains the worst PPL and TC scores, which can be attributed to its tendency to extract frequently occurring themes from training, hindering its fast adaptability to meta-test sets with only several documents. *ii)* To capture the relations of texts and model the task information, in Section 2.2, we posit a context-specific variable $c^{(i)}$ as the prior of document-level topic proportion $\theta_j^{(i)}$. It can be found with $c^{(i)}$, our Meta-CETM achieves much lower PPL and higher TC, demonstrating the efficacy of task-specific variables. *iii)* For ablation results without the **Graph VAE** module, we replace it with a **Graph AE** rather than discarding it completely. Besides, for Meta-CETM without the **GMM prior** design, we replace it with the **standard Gaussian distribution prior** rather than imposing no prior completely. From the quantitative results shown

Table 3: Ablation study on four datasets. The number of texts in each task is 10. "✓" means we add the corresponding design into ETM [18], which is chosen as the baseline.

| | context variable | Graph VAE | GMM prior | 20NG | | | DB14 | | |
|---|---|---|---|---|---|---|---|---|---|
| | | | | PPL | TD | TC | PPL | TD | TC |
| ETM | | | | 3107 | **0.8395** | -0.8437 | 3054 | **0.8106** | -0.8719 |
| | | ✓ | ✓ | 1964 | 0.8031 | -0.4301 | 1682 | 0.7441 | -0.4917 |
| | ✓ | | ✓ | 1255 | 0.7983 | -0.4169 | 1131 | 0.7210 | -0.5025 |
| | ✓ | ✓ | | 1361 | 0.6538 | -0.4688 | 1276 | 0.6562 | -0.6677 |
| **Meta-CETM** | ✓ | ✓ | ✓ | **1170** | 0.8154 | **-0.3701** | **1084** | 0.7475 | **-0.4783** |
| | context variable | Graph VAE | GMM prior | Yahoo | | | WOS | | |
| | | | | PPL | TD | TC | PPL | TD | TC |
| ETM | | | | 2817 | **0.8851** | -0.8913 | 3310 | **0.9286** | -0.9785 |
| | | ✓ | ✓ | 1906 | 0.7243 | -0.5097 | 2023 | 0.9141 | -0.5420 |
| | ✓ | | ✓ | 1316 | 0.7612 | -0.4860 | 1304 | 0.8387 | -0.5262 |
| | ✓ | ✓ | | 1271 | 0.5847 | -0.5503 | 1389 | 0.7018 | -0.5587 |
| **Meta-CETM** | ✓ | ✓ | ✓ | **1219** | 0.7886 | **-0.4639** | **1293** | 0.8667 | **-0.5177** |

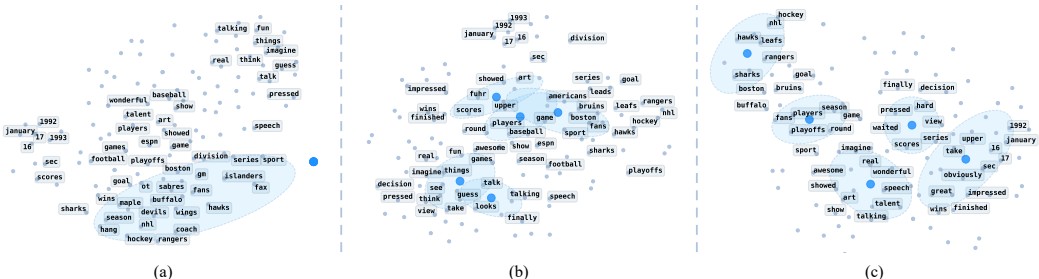

(a)               (b)               (c)

Figure 5: Illustration of the advantage of using a GMM prior. The adapted embedding space of our model by using (a) no prior, *i.e.,* Graph VAE is replaced with a vanilla graph AE, and the topic embeddings are learnable point vectors, (b) a standard normal prior, *i.e.,* the topic embeddings are learnable Gaussian distributions which are constrained by KL divergence with the standard normal distribution, (c) a GMM prior. The target task is sampled from the sub-topic "rec.sport.hockey" of the 20NG dataset. Room in for better visual effects.

in the table, it can be found compared with Gaussian prior in vanilla VAE model, GMM priors encourage the semantically coherent words to be allocated to the same topic, leading to higher TC scores and lower PPL. Further, we visualize the embedding space in Figure 5 to demonstrate the benefits of GMM prior.

## 4   Related Work

To discover a group of topics from a batch of documents, probabilistic topic models have been developed in recent years, including hierarchical topic models [54, 14, 46, 55], NTMs [56, 15], embedded topic models [18, 31], and CTMs [22, 21]. Besides, some works are proposed to incorporate knowledge into the modeling process [57–59] and some researchers apply optimal transport to measure the distances between topics and words or documents [60, 61]. From the perspective of modelling word embeddings with Gaussian distribution, Vilnis *etc.* proposed a novel density-based mapping method [62] and achieved promising performances. Recently, some works [30, 27] aimed at topic modelling under the few-shot setting and proposed to obtain a group of task embeddings to adaptively explore topics within only a few documents in a model-based meta-learning fashion. However, their method cannot address the multiple meanings of one word issue, a prevalent issue in practical document analysis, which is addressed through the introduction of dependency graph and the GMM prior distribution in our work.

## 5   Conclusion

In this paper, we propose a novel NTM, Meta-CETM, to address the the fast adaption problem in document analysis under low-resource regimes. Specifically, we construct a task-specific graph to obtain context-related word embeddings. Then we introduce the graph VAE with Gaussian mixture prior to model the word representations and topic embeddings, which are optimized through the EM algorithm. We also propose the task-specific prior for topic proportions. Through extensive experiments and illustrations, we demonstrate the superior performance of our model in solving the adaptation problem in topic modeling.

## 6   Acknowledgements

This work was supported in part by the National Natural Science Foundation of China under Grant U21B2006; in part by Shaanxi Youth Innovation Team Project; in part by the Fundamental Research Funds for the Central Universities QTZX23037 and QTZX22160; in part by the 111 Project under Grant B18039; in part by the Fundamental Research Funds for the Central Universities; in part by the Innovation Fund of Xidian University under Grant YJSJ23016.

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

# A Discussions

## A.1 Limitations and Future Work

In this paper, we propose a method to improve existing embedded topic models (ETMs) under the low-resource settings by introducing the task-specific graph with the Gaussian mixture prior. The main limitation of our work could be the reliance on the pre-trained parsing tools. Specifically, we construct the semantic graph according to the syntactic structure of the documents with the public library, spaCy[7]. However, some other approaches such as the sliding window method can also be used to construct the graph, which is not relied on training. Besides, although strong performances, our framework is still trained and evaluated on the same dataset. While, in the current large models era, pre-training on a large set and then perform few-shot learning or even zero-shot learning attracts researchers' attention and interest, especially such Bayesian models. However, this is beyond the scope of this paper and we will conduct a thorough investigation of this issue in future work.

## A.2 Broader impact

Our work builds a novel and advanced topic modeling technique under few-shot settings. For example, at the very beginning of the Covid-19, there were relatively few cases of infection in all regions, and the reports of related medical diagnoses were highly limited. Under such circumstances, few-shot topic models can help us extract key information from limited resources, thus facilitating us to take appropriate and efficient preventive and control measures. On the other hand, the emergence of ChatGPT by OpenAI has attracted a great deal of attention, while in this paper, what we explore is a robust model with generalization capabilities given limited text data, rather than training from a very large corpus. Potential negative societal impact of our work could arise from malicious intent in changing model's behavior by injecting deliberate human prejudice, which may harm the fairness of the community. However, we hope our work is utilized to enable more research and applications primarily from the originality of benefiting the community development.

# B Key Notations

In Table 4, we list the key notations, descriptions and corresponding dimensions used in this paper.

Table 4: Notations used in the paper.

| Symbol | Dimensionality | Description |
|---|---|---|
| $M$ | - | number of total training tasks |
| $J$ | - | number of documents in each task |
| $K$ | - | number of topics in each task |
| $V$ | - | number of vocabulary terms, shared across tasks |
| $D$ | - | dimensionality of the word latent space |
| $\mathcal{T}^{(i)}$ | - | the $i^{th}$ training task |
| $\mathbf{X}^{(i)}$ | $\mathbb{R}^{V \times J}$ | the BoWs representations for documents in the $i^{th}$ task |
| $\mathbf{H}^{(i)}$ | $\mathbb{R}^{300 \times J}$ | the deterministic hidden features of BoWs $\mathbf{X}^{(i)}$ |
| $\boldsymbol{c}^{(i)}$ | $\mathbb{R}^{K}$ | context variable that summarizes the topic proportion information |
| $\boldsymbol{\theta}_j^{(i)}$ | $\mathbb{R}^{K}$ | topic proportion of the $j^{th}$ in the $i^{th}$ task |
| $\boldsymbol{\beta}^{(i)}$ | $\mathbb{R}^{V \times K}$ | topic-word matrix for the $i^{th}$ task |
| $\mathbf{A}^{(i)}$ | $\mathbb{R}^{V \times V}$ | the adjacency matrix of dependency graph for the $i^{th}$ task |
| $\mathbf{e}_v^{(i)}$ | $\mathbb{R}^{D}$ | initialized features of the $v^{th}$ word appeared in the $i^{th}$ task |
| $\mathbf{z}_v^{(i)}$ | $\mathbb{R}^{D}$ | adaptive embedding of the $v^{th}$ word appeared in the $i^{th}$ task |
| $\pi_k^{(i)}$ | - | coefficient of the $k^{th}$ Gaussian component for the $i^{th}$ task |
| $\boldsymbol{\mu}_k^{(i)}$ | $\mathbb{R}^{D}$ | mean of the $k^{th}$ Gaussian component for the $i^{th}$ task |
| $\boldsymbol{\Sigma}_k^{(i)}$ | $\mathbb{R}^{D \times D}$ | covariance of the $k^{th}$ Gaussian component for the $i^{th}$ task |

---

[7]https://spacy.io/

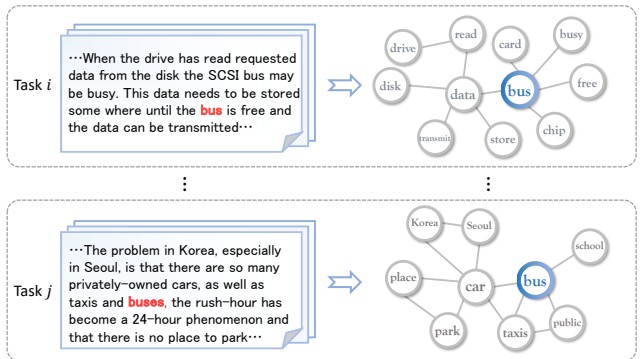

Figure 6: An illustration of word sense variation caused by different contexts. The task $i$ is sampled from a corpus about "hardware", and the task $j$ is sampled from a corpus related to "autos".

## C   An Illustration of Our Settings

In Section 2.2, we touched on the concepts of corpus, task, and document to describe our problem setting, which is a bit messy to follow. Here, we clarify these concepts through a concrete example so that the reader could understand the problem setting of few-shot learning more easily. Considering the 20NG [38] dataset:

A "**corpus**" refers to a collection of documents belonging to the same category so that 20NG consists of 20 corpora, each of which contains documents from one of the 20 classes.

A "**task**" is a smaller unit than a "corpus" that only comprises a few related documents, typically less than 100 documents. Consequently, we could sample a number of tasks from each training corpus.

(we select 12 out of the 20 corpora for training)

Then our goal is to utilize these sampled tasks to train a generalizable topic model that can efficiently adapt to a new task from the test corpus (the remaining 8 corpora are used for testing). In addition, for each task at the testing stage, we split its documents into two parts, one for fine-tuning or retraining the topic model, called the **support set**, and the other for evaluating the model's performance, called the **query set**. Note that we do not design different generative processes for the corpus documents versus the task documents. In essence, our proposed Meta-CETM only characterizes the generative process of the task documents by jointly modeling the syntactic graph $\mathbf{A}$ and the observed BoW $\mathbf{X}$ in each task. In Fig. 6, we visualize the task and the corresponding unweighted dependency graph $\mathbf{A}$.

## D   Algorithms for training and testing

In Alg. 1 and Alg. 2, we present the training and meta-testing procedures of our Meta-CETM.

## E   Derivation of Formulas

In this section, we provide the detailed derivation process of variational evidence lower bound (ELBO) in Eq. 8 and the expectation maximization solver process for multivariate Gaussian distribution in Eq. 7 in Sec. 2.2.

**Algorithm 1:** Training process

**Input:** A set of training corpora $\{\mathcal{D}_c\}_{c=1}^C$; initialized model parameters $\Psi$

Randomly sample tasks from each training corpus $\mathcal{D}_c$ to obtain $\{\mathcal{T}^{(i)}\}_{i=1}^M$;

**for** *each task* $\mathcal{T}^{(i)}, i = 1, 2, \cdots, M$ **do**

    Build semantic graph $\mathbf{A}^{(i)}$ with established dependency parsing tools;

    Infer adaptive word embeddings $\mathbf{Z}^{(i)}$ according to Eq. 6;

    Initialize parameters of the Gaussian mixture prior: $\pi_k$, $\boldsymbol{\mu}_k$ and $\boldsymbol{\Sigma}_k$;

    Update to the optimal value $\pi_k^{(i)}$, $\boldsymbol{\mu}_k^{(i)}$ and $\boldsymbol{\Sigma}_k^{(i)}$ using EM based on Eq. 7;

    Compute the topic-word matrix $\boldsymbol{\beta}^{(i)}$ according to Eq. 3;

    Infer the latent context varibale $\boldsymbol{c}^{(i)}$ using Eq. 5;

    **for** *each document* $\mathbf{x}_j^{(i)}, j = 1, 2, \cdots, J$ **do**

        Infer topic proportion $\boldsymbol{\theta}_j^{(i)}$ with Eq. 4;

        Calculate the log-likelihood $p(\mathbf{x}_j^{(i)}|\boldsymbol{\theta}_j^{(i)}, \boldsymbol{\beta}^{(i)})$;

    Derive the ELBO as Eq. 8 and update $\Psi$ using SGD;

---

**Algorithm 2:** Meta-test for a new task

**Input:** A new corpus $\mathcal{D}_{test}$, trained model parameters $\Psi$

**Output:** Adaptive topic-word matrix $\boldsymbol{\beta}$

Randomly sample a task $\mathcal{T}_{new}$ from the given corpus $\mathcal{D}_{test}$;

Get the corresponding BoWs $\mathbf{X}_{new}$ and dependency graph $\mathbf{A}_{new}$ for the current task;

Infer the adaptive word embeddings $\mathbf{Z}_{new}$ with part of the trained model parameters $\Psi$;

Initialize parameters of the Gaussian mixture prior: $\pi_k$, $\boldsymbol{\mu}_k$ and $\boldsymbol{\Sigma}_k$;

Compute optimal $\pi_k^*$, $\boldsymbol{\mu}_k^*$ and $\boldsymbol{\Sigma}_k^*$ using EM based on Eq. 7;

Derive the adaptive topic-word matrix $\boldsymbol{\beta}_{new}$ by Eq. 3;

---

### E.1 Variational ELBO

$$
\begin{aligned}
\log p(X^{(i)}, A^{(i)}) &= \log \iiint p(X^{(i)}, A^{(i)}, \Theta^{(i)}, c^{(i)}, Z^{(i)}) d\Theta^{(i)} dc^{(i)} dZ^{(i)} \\
&= \log \iiint p(X^{(i)} \mid \Theta^{(i)}, Z^{(i)}) p(\Theta^{(i)} \mid c^{(i)}) p(c^{(i)}) p(A^{(i)} \mid Z^{(i)}) p(Z^{(i)}) d\Theta^{(i)} dc^{(i)} dZ^{(i)} \\
&= \log \mathbb{E}_Q \left[ \frac{p(X^{(i)} \mid \Theta^{(i)}, Z^{(i)}) p(\Theta^{(i)} \mid c^{(i)}) p(c^{(i)}) p(A^{(i)} \mid Z^{(i)}) p(Z^{(i)})}{q(\Theta^{(i)} \mid X^{(i)}, c^{(i)}) q(c^{(i)} \mid X^{(i)}) q(Z^{(i)} \mid A^{(i)}, E^{(i)})} \right] \\
&\geq \mathbb{E}_Q \left[ \log \frac{p(X^{(i)} \mid \Theta^{(i)}, Z^{(i)}) p(\Theta^{(i)} \mid c^{(i)}) p(c^{(i)}) p(A^{(i)} \mid Z^{(i)}) p(Z^{(i)})}{q(\Theta^{(i)} \mid X^{(i)}, c^{(i)}) q(c^{(i)} \mid X^{(i)}) q(Z^{(i)} \mid A^{(i)}, E^{(i)})} \right] \\
&= \mathbb{E}_Q \left[ \log \prod_{j=1}^J p(x_j^{(i)} \mid \theta_j^{(i)}, Z^{(i)}) \right] + \mathbb{E}_Q \left[ \log \prod_{j=1}^J \frac{p(\theta_j^{(i)} \mid c^{(i)})}{q(\theta_j^{(i)} \mid x_j^{(i)}, c^{(i)})} \right] \\
&\quad + \mathbb{E}_Q \left[ \log \frac{p(c^{(i)})}{q(c^{(i)} \mid X^{(i)})} \right] + \mathbb{E}_Q \left[ \log p(A^{(i)} \mid Z^{(i)}) \right] + \mathbb{E}_Q \left[ \log \frac{p(Z^{(i)})}{q(Z^{(i)} \mid A^{(i)}, E^{(i)})} \right] \\
&= \sum_{j=1}^J \mathbb{E}_Q \left[ \log p(x_j^{(i)} \mid \theta_j^{(i)}, Z^{(i)}) \right] + \sum_{j=1}^J \mathbb{E}_Q \left[ \log \frac{p(\theta_j^{(i)} \mid c^{(i)})}{q(\theta_j^{(i)} \mid x_j^{(i)}, c^{(i)})} \right] \\
&\quad + \mathbb{E}_Q \left[ \log \frac{p(c^{(i)})}{q(c^{(i)} \mid X^{(i)})} \right] + \mathbb{E}_Q \left[ \log p(A^{(i)} \mid Z^{(i)}) \right] + \mathbb{E}_Q \left[ \log \frac{p(Z^{(i)})}{q(Z^{(i)} \mid A^{(i)}, E^{(i)})} \right] \\
&= \mathcal{L}_{ELBO}
\end{aligned}
$$

$$(11)$$

## E.2 Solving topic parameters $\{\pi_k^{(i)}, \mu_k^{(i)}, \Sigma_k^{(i)}\}_{k=1}^K$ with Expectation Maximization

The log likelihood function is given by

$$\ln p(Z^{(i)} \mid \pi^{(i)}, \mu^{(i)}, \Sigma^{(i)}) = \sum_{v=1}^V \ln \left[ \sum_{k=1}^K \pi_k^{(i)} \mathcal{N}(z_v^{(i)} \mid \mu_k^{(i)}, \Sigma_k^{(i)}) \right]. \tag{12}$$

**1. Deriving $\mu_k^{(i)}$**

Setting the derivatives of $\ln p(Z^{(i)} \mid \pi^{(i)}, \mu^{(i)}, \Sigma^{(i)})$ w.r.t the means $\mu_k^{(i)}$ to zero, we have

$$-\sum_{v=1}^V \frac{\pi_k^{(i)} \mathcal{N}(z_v^{(i)} \mid \mu_k^{(i)}, \Sigma_k^{(i)})}{\sum_{s=1}^K \pi_s^{(i)} \mathcal{N}(z_v^{(i)} \mid \mu_s^{(i)}, \Sigma_s^{(i)})} \Sigma_k^{(i)}(z_v^{(i)} - \mu_k^{(i)}) = 0. \tag{13}$$

Define the posterior probabilities as

$$\gamma_{vk} = p(y_v^{(i)} = k \mid z_v^{(i)}) = \frac{\pi_k^{(i)} \mathcal{N}(z_v^{(i)} \mid \mu_k^{(i)}, \Sigma_k^{(i)})}{\sum_{s=1}^K \pi_s^{(i)} \mathcal{N}(z_v^{(i)} \mid \mu_s^{(i)}, \Sigma_s^{(i)})}. \tag{14}$$

Multiplying by $\Sigma_k^{(i)^{-1}}$ and rearranging, we can obtain the updating formula for $\mu_k^{(i)}$ as

$$\mu_k^{(i)} = \frac{\sum_v \gamma_{vk} \cdot z_v^{(i)}}{\sum_v \gamma_{vk}}. \tag{15}$$

**2. Deriving $\Sigma_k^{(i)}$**

Similarly, we set the derivatives of $\ln p(Z^{(i)} \mid \pi^{(i)}, \mu^{(i)}, \Sigma^{(i)})$ w.r.t $\Sigma_k^{(i)}$ to zero, then we have

$$-\frac{1}{2} \sum_{v=1}^V \frac{\pi_k^{(i)} \mathcal{N}(z_v^{(i)} \mid \mu_k^{(i)}, \Sigma_k^{(i)})}{\sum_{s=1}^K \pi_s^{(i)} \mathcal{N}(z_v^{(i)} \mid \mu_s^{(i)}, \Sigma_s^{(i)})} \Sigma_k^{(i)^{-1}} \left[ 1 + (z_v^{(i)} - \mu_k^{(i)})^T \Sigma_k^{(i)^{-1}} (z_v^{(i)} - \mu_k^{(i)}) \right] = 0. \tag{16}$$

Using $\gamma_{vk}$ in Eq. 4 and rearranging, we get the updating formula for $\Sigma_k^{(i)}$ as

$$\Sigma_k^{(i)} = \frac{\sum_v \gamma_{vk} \cdot (z_v^{(i)} - \mu_k^{(i)})(z_v^{(i)} - \mu_k^{(i)})^T}{\sum_v \gamma_{vk}}. \tag{17}$$

**3. Deriving $\pi_k^{(i)}$**

Finally, using Lagrange multiplier algorithm, our goal is to maximize the following formula:

$$\sum_{v=1}^V \ln \left[ \sum_{k=1}^K \pi_k^{(i)} \mathcal{N}(z_v^{(i)} \mid \mu_k^{(i)}, \Sigma_k^{(i)}) \right] + \lambda(\sum_{k=1}^K \pi_k^{(i)} - 1), \tag{18}$$

where $\sum_{k=1}^K \pi_k^{(i)} = 1$.

Then setting the derivatives of the above equation w.r.t $\pi_k^{(i)}$ to zero, we have

$$\sum_{v=1}^V \frac{\pi_k^{(i)} \mathcal{N}(z_v^{(i)} \mid \mu_k^{(i)}, \Sigma_k^{(i)})}{\sum_{s=1}^K \pi_s^{(i)} \mathcal{N}(z_v^{(i)} \mid \mu_s^{(i)}, \Sigma_s^{(i)})} + \lambda = 0. \tag{19}$$

Multiplying $\pi_k^{(i)}$ and rearranging, we obtain

$$\pi_k^{(i)} = -\frac{\sum_{v=1}^V \frac{\pi_k^{(i)} \mathcal{N}(z_v^{(i)} \mid \mu_k^{(i)}, \Sigma_k^{(i)})}{\sum_{s=1}^K \pi_s^{(i)} \mathcal{N}(z_v^{(i)} \mid \mu_s^{(i)}, \Sigma_s^{(i)})}}{\lambda} = -\frac{\sum_v \gamma_{vk}}{\lambda}. \tag{20}$$

Considering $\sum_{k=1}^K \pi_k^{(i)} = 1$, then $\sum_k -\frac{\sum_v \gamma_{vk}}{\lambda} = 1$, and $\lambda = \sum_v \sum_k \gamma_{vk}$.

Hence the updating formula for $\pi_k^{(i)}$ as

$$\pi_k^{(i)} = \frac{\sum_v \gamma_{vk}}{\sum_v \sum_k \gamma_{vk}}. \tag{21}$$

Table 5: PPL results on four datasets with 20 documents in each task.

| Methods | 20NG | Yahoo | DB14 | WOS |
|---------|------|-------|------|-----|
| LDA[42] | 2979 | 3916 | 3095 | 2370 |
| PFA[12] | 2439 | 2545 | 1903 | 1675 |
| ProdLDA[43] | 4807 | 6093 | 5819 | 4617 |
| ETM[18] | 3276 | 2781 | 2870 | 3189 |
| MAML-ProdLDA* | 4378 | 4033 | 4612 | 3908 |
| MAML-ETM* | 3287 | 3439 | 2819 | 4189 |
| Meta-SawETM[30] | 2657 | 2859 | 2355 | 3620 |
| CombinedTM[21] | 2331 | 2543 | 1863 | 2587 |
| ZeroShotTM[22] | 2673 | 2664 | 1722 | 2660 |
| **Meta-CETM** | **1216** | **1369** | **1109** | **1482** |

# F More Results

## F.1 Perplexity results

In Table 1, we list the PPL results of different compared methods on four datasets, where the number of documents is 5 or 10 in each task. In this part, we provide more PPL quantitative results with varied numbers of texts per task {20, 50, 100} in Table 5, Table 6 and Table 7, respectively.

Table 6: PPL results on four datasets with 50 documents in each task.

| Methods | 20NG | Yahoo | DB14 | WOS |
|---------|------|-------|------|-----|
| LDA[42] | 2443 | 3279 | 2353 | 2091 |
| PFA[12] | 2271 | 2326 | 1887 | 1663 |
| ProdLDA[43] | 4489 | 5784 | 5794 | 4386 |
| ETM[18] | 3215 | 3916 | 3095 | 2370 |
| MAML-ProdLDA* | 4372 | 3951 | 4463 | 3863 |
| MAML-ETM* | 3186 | 3315 | 2778 | 4062 |
| Meta-SawETM[30] | 3761 | 3037 | 2577 | 3365 |
| CombinedTM[21] | 2267 | 2481 | 1765 | 2473 |
| ZeroShotTM[22] | 2397 | 2496 | 1638 | 2497 |
| **Meta-CETM** | **1517** | **1440** | **1306** | **1576** |

## F.2 Topic quality results

In Sec. 3.2.2, we present the topic interpretability results including topic diversity (TD) and topic coherence (TC) of six compared methods. Except for CombinedTM [21] and ZeroShotTM [22], we carry on experiments applying another contextual topic model (CTM) CETopicTM [24] with SimCSE pretrained word embeddings[8] on four datasets. The results are exhibited in Fig. 7. It can be notably noticed CETopicTM [24] achieves much competitive results on both TD and TC scores, even compared with CombinedTM [21] and ZeroShotTM [22]. Such superiority is owed to the fact that CETopicTM utilizes word embeddings learned from large-scale BERT data and it performs clustering on sentence embeddings to generate topics. In our settings, the aim is to provide a framework for training a sufficiently generalized topic model in low-resource regimes, while equipped with BERT embeddings, CETopicTM is highly likely to obtain context-related meanings in advance under most situations. But in some cases where the words or the word meanings have not been encountered or

---

[8]https://huggingface.co/princeton-nlp/unsup-simcse-bert-base-uncased

Table 7: PPL results on four datasets with 100 documents in each task.

| Methods | 20NG | Yahoo | DB14 | WOS |
|---|---|---|---|---|
| LDA[42] | 2118 | 2833 | 1858 | 1896 |
| PFA[12] | **2060** | 2169 | 1637 | **1643** |
| ProdLDA[43] | 4466 | 5736 | 6016 | 4369 |
| ETM[18] | 3199 | 2811 | 2837 | 3296 |
| MAML-ProdLDA* | 4359 | 4202 | 4381 | 3845 |
| MAML-ETM* | 3172 | 3256 | 2715 | 3947 |
| Meta-SawETM[30] | 3661 | 3251 | 2984 | 3101 |
| CombinedTM[21] | 2205 | 2286 | 1695 | 2330 |
| ZeroShotTM[22] | 2330 | 2319 | 1604 | 2372 |
| **Meta-CETM** | 2138 | **1743** | **1468** | 1786 |

learned by BERT, such as some specialized occasions, CETopicTM may fail to extract interpretable topics.

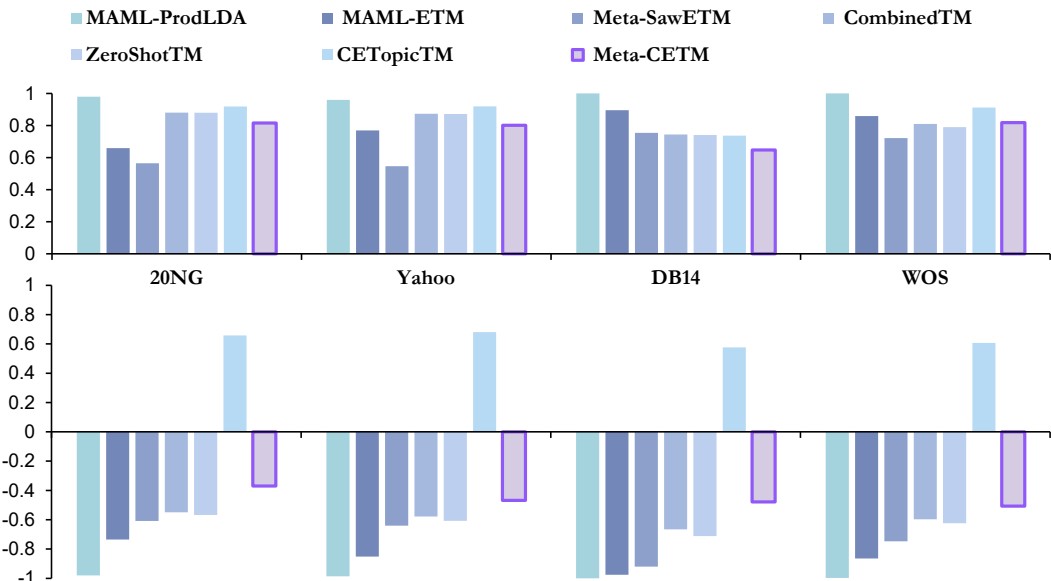

Figure 7: Topic diversity results (top row) and topic coherence results (bottom row) of seven compared methods on four datasets. Compared with Fig. 2, we add the results of CETopicTM [24] in this figure.

## F.3 Topic visualization results

In Fig. 3, we visualize the adapted embedding space of different methods to demonstrate our Meta-CETM's successful fast adaption. Further, to better characterize meaningful and coherent topics learned by our model given a few number of documents, we display the text and topics extracted by Meta-SawETM [30], CombinedTM [21] and our Meta-CETM in Fig. 8.

… **What sports would you say it is easy and difficult to be a rookie in? The sports that I would say it is difficult to be a rookie in are basketball, hockey, and soccer. The easiest sport for a rookie is certainly the NFL. Some of the greatest NFL players started out on special teams. They have put much efforts to help the team win the game, especially the super bowl** …

| Meta-SawETM | | |
|---|---|---|
| tv | music | school |
| air | like | help |
| ball | teams | nfl |
| players | time | sports |
| team | player | people |

| CombinedTM | | |
|---|---|---|
| champion | play | players |
| weekend | ice | team |
| perfect | friends | working |
| fort | effort | believe |
| hockey | nfl | breaks |

| Meta-CETM | | |
|---|---|---|
| basketball | soccer | rookie |
| nfl | nfl | nfl |
| baseball | sport | team |
| greatest | rookie | game |
| putting | super | win |

Figure 8: A paragraph of text and top five words of three topics from Meta-SawETM, CombinedTM and our Meta-CETM. It can be clearly found that Meta-CETM learns the most relevant topics among the three models.

