# Supplementary Materials for "Context-guided Embedding Adaptation for Effective Topic Modeling in Low-Resource Regimes"

## 1 Key Notations

In Table 1, we list the key notations, descriptions and corresponding dimensions used in this paper.

Table 1: Notations used in the paper.

| Symbol | Dimensionality | Description |
|---|---|---|
| $M$ | - | number of total training tasks |
| $J$ | - | number of documents in each task |
| $K$ | - | number of topics in each task |
| $V$ | - | number of vocabulary terms, shared across tasks |
| $D$ | - | dimensionality of the word latent space |
| $\mathcal{T}^{(i)}$ | - | the $i^{th}$ training task |
| $\mathbf{X}^{(i)}$ | $\mathbb{R}^{V \times J}$ | the BoWs representations for documents in the $i^{th}$ task |
| $\mathbf{H}^{(i)}$ | $\mathbb{R}^{300 \times J}$ | the deterministic hidden features of BoWs $\mathbf{X}^{(i)}$ |
| $\boldsymbol{c}^{(i)}$ | $\mathbb{R}^{K}$ | context variable that summarizes the topic proportion information |
| $\boldsymbol{\theta}_j^{(i)}$ | $\mathbb{R}^{K}$ | topic proportion of the $j^{th}$ in the $i^{th}$ task |
| $\boldsymbol{\beta}^{(i)}$ | $\mathbb{R}^{V \times K}$ | topic-word matrix for the $i^{th}$ task |
| $\mathbf{A}^{(i)}$ | $\mathbb{R}^{V \times V}$ | the adjacency matrix of dependency graph for the $i^{th}$ task |
| $\mathbf{e}_v^{(i)}$ | $\mathbb{R}^{D}$ | initialized features of the $v^{th}$ word appeared in the $i^{th}$ task |
| $\mathbf{z}_v^{(i)}$ | $\mathbb{R}^{D}$ | adaptive embedding of the $v^{th}$ word appeared in the $i^{th}$ task |
| $\pi_k^{(i)}$ | - | coefficient of the $k^{th}$ Gaussian component for the $i^{th}$ task |
| $\boldsymbol{\mu}_k^{(i)}$ | $\mathbb{R}^{D}$ | mean of the $k^{th}$ Gaussian component for the $i^{th}$ task |
| $\boldsymbol{\Sigma}_k^{(i)}$ | $\mathbb{R}^{D \times D}$ | covariance of the $k^{th}$ Gaussian component for the $i^{th}$ task |

## 2 Algorithms for training and testing

In this section, we present the training and meta-testing procedures of our Meta-CETM in Alg. 1 and Alg. 2, respectively.

Submitted to 37th Conference on Neural Information Processing Systems (NeurIPS 2023). Do not distribute.

---

**Algorithm 1:** Training process

---

**Input:** A set of training corpora $\{\mathcal{D}_c\}_{c=1}^{C}$; initialized model parameters $\Psi$
Randomly sample tasks from each training corpus $\mathcal{D}_c$ to obtain $\{\mathcal{T}^{(i)}\}_{i=1}^{M}$;
**for** *each task* $\mathcal{T}^{(i)}, i = 1, 2, \cdots, M$ **do**
    Build semantic graph $\mathbf{A}^{(i)}$ with established dependency parsing tools;
    Infer adaptive word embeddings $\mathbf{Z}^{(i)}$ according to Eq. 6;
    Initialize parameters of the Gaussian mixture prior: $\pi_k$, $\boldsymbol{\mu}_k$ and $\boldsymbol{\Sigma}_k$;
    Update to the optimal value $\pi_k^{(i)}$, $\boldsymbol{\mu}_k^{(i)}$ and $\boldsymbol{\Sigma}_k^{(i)}$ using EM based on Eq. 7;
    Compute the topic-word matrix $\boldsymbol{\beta}^{(i)}$ according to Eq. 3;
    Infer the latent context varibale $\boldsymbol{c}^{(i)}$ using Eq. 5;
    **for** *each document* $\mathbf{x}_j^{(i)}, j = 1, 2, \cdots, J$ **do**
        Infer topic proportion $\boldsymbol{\theta}_j^{(i)}$ with Eq. 4;
        Calculate the log-likelihood $p(\mathbf{x}_j^{(i)}|\boldsymbol{\theta}_j^{(i)}, \boldsymbol{\beta}^{(i)})$;
    Derive the ELBO as Eq. 8 and update $\Psi$ using SGD;

---

---

**Algorithm 2:** Meta-test for a new task

---

**Input:** A new corpus $\mathcal{D}_{test}$, trained model parameters $\Psi$
**Output:** Adaptive topic-word matrix $\boldsymbol{\beta}$
Randomly sample a task $\mathcal{T}_{new}$ from the given corpus $\mathcal{D}_{test}$;
Get the corresponding BoWs $\mathbf{X}_{new}$ and dependency graph $\mathbf{A}_{new}$ for the current task;
Infer the adaptive word embeddings $\mathbf{Z}_{new}$ with part of the trained model parameters $\Psi$;
Initialize parameters of the Gaussian mixture prior: $\pi_k$, $\boldsymbol{\mu}_k$ and $\boldsymbol{\Sigma}_k$;
Compute optimal $\pi_k^*$, $\boldsymbol{\mu}_k^*$ and $\boldsymbol{\Sigma}_k^*$ using EM based on Eq. 7;
Derive the adaptive topic-word matrix $\boldsymbol{\beta}_{new}$ by Eq. 3;

---

## 3 An Illustration of Our Settings

In the main paper, we mention corpus, task, document, support set and query set to present our framework, which is a bit messy to follow. Here, we provide a clarification of these mechanics following the literature in few-shot learning problems for better understanding.

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

## 5.3 Few-shot document classification results

In main paper, we list the classification results without intervals in Table.2 in terms of the space limit. In this section, we provide the complete results of different compared methods with confidence intervals.

Table 2: 5-way 5-shot and 5-way 10-shot few-shot text classification results with intervals. * denotes all parameters of the model are fine-tuned.

| Methods | | 20NG | | DB14 | |
|---------|---------|--------------|---------------|--------------|---------------|
| Rep. | Alg. | 5 shot | 10 shot | 5 shot | 10 shot |
| MLP | MAML | $32.01 \pm 0.53$ | $36.20 \pm 0.21$ | $50.20 \pm 1.28$ | $60.30 \pm 0.85$ |
| | PROTO | $35.20 \pm 0.66$ | $38.30 \pm 0.45$ | $54.13 \pm 0.89$ | $57.16 \pm 0.72$ |
| | FT | $29.70 \pm 0.75$ | $33.04 \pm 0.57$ | $51.11 \pm 1.82$ | $53.83 \pm 1.74$ |
| | FT* | $38.87 \pm 0.51$ | $48.52 \pm 0.34$ | $71.12 \pm 1.04$ | $77.94 \pm 0.76$ |
| CNN | MAML | $34.08 \pm 0.41$ | $45.40 \pm 1.51$ | $66.28 \pm 1.07$ | $75.96 \pm 0.98$ |
| | PROTO | $39.86 \pm 0.79$ | $49.71 \pm 0.62$ | $\mathbf{78.58} \pm 0.90$ | $\mathbf{81.01} \pm 0.65$ |
| | FT | $\underline{45.70} \pm 0.47$ | $\underline{53.63} \pm 0.29$ | $74.68 \pm 1.58$ | $\underline{80.75} \pm 0.96$ |
| | FT* | $44.53 \pm 0.71$ | $51.92 \pm 0.39$ | $72.49 \pm 1.64$ | $80.07 \pm 1.29$ |
| HNS-SawETM | | $39.37 \pm 0.78$ | $43.78 \pm 0.93$ | $65.93 \pm 1.15$ | $71.08 \pm 0.67$ |
| Meta-SawETM | | $39.19 \pm 0.95$ | $45.83 \pm 0.75$ | $67.20 \pm 1.53$ | $72.31 \pm 1.33$ |
| CombinedTM | | $46.17 \pm 0.94$ | $52.73 \pm 0.69$ | $68.42 \pm 1.19$ | $73.26 \pm 1.03$ |
| ZeroShotTM | | $46.65 \pm 0.59$ | $52.08 \pm 0.53$ | $71.93 \pm 1.74$ | $76.09 \pm 1.23$ |
| Meta-CETM | | $\mathbf{50.57} \pm 0.27$ | $\mathbf{58.47} \pm 0.14$ | $\underline{76.85} \pm 1.37$ | $79.34 \pm 1.18$ |