# OpenReview forum: "Context-guided Embedding Adaptation for Effective Topic Modeling in Low-Resource Regimes"
_NeurIPS.cc/2023/Conference — NeurIPS 2023 poster_

### Official Review · Reviewer_CvqN · 2023-06-13

**Soundness:** 3 good
**Presentation:** 3 good
**Contribution:** 3 good
**Rating:** 7
**Confidence:** 4

**Summary:**

This paper proposes a new solution (Meta-CETM) for inferring topics on a dataset with a few available documents only. The main idea is to train the model on various tasks and then use it on a new, small dataset. The are extensive experiments on diverse topic models, in particular in this context of "few shot" learning, and the results show that Meta-CETM outperforms its competitors.

**Strengths:**

- few-shot learning for topic modeling is an important topic today
- the model looks sound (even though not really well explained)
- the experimental framework is quite strong, with good results for the proposed solution


**Weaknesses:**

- The solution is not always well presented
- It's a quite complicated model with many intertwined modules

**Questions:**

I have some concerns on this paper.

- The paper is not always well presented. Here I give some examples:
  * The problem formulation (2.1) is quite confusing, probably because of the limited space. This is much clearer in the supplementary material. Some aspects are not sufficiently described or motivated. For instance, the authors seem to use a self attention mechanism similar to the one in the Transformer (noted Attn). However, this mechanism is normally based on an embedding matrix, which is not the case here for X is a simple BoW representation of the documents.
  * Why using the Weibull distribution in this context? There is no explanation at all.
  * What "latent indication" means for H(i)?
  * I guess the step 1.c line 114 is done after having computed all the word embedding of the generated document (we need all the Z(i)). Is it true? If so, this step should be taken outside the for loop.

- Would it be possible to use the adaptive word embedding module (Z(i)) for the bag-of-word encoder?

- The experimental part of the paper looks quite strong. Important solutions for topic modeling (old and new) are considered in the competitors and the authors use several manners to compete with them (perplexity, classification, topic coherence and diversity, qualitative evaluation). However, I would expect more experiments on the few-shot context, for instance by varying the number of available documents (not only 5 or 10) and taking into account the semantic difference between the corpora (i.e., between the training set and the test set).

**Limitations:**

There is not a single limitation mentioned in the paper... There is no future work as well.

---

> ### Author Rebuttal · Authors · 2023-08-08
>
> Thank you for your positive and helpful comments and suggestions. Your concerns have been addressed as follows.
>
>
> **Q1:** The paper is not always well presented. Here I give some examples:
>
> - *Q1.1:* The problem formulation (2.1) is quite confusing ... For instance, the authors seem to use a self-attention mechanism similar to the one in the Transformer (noted Attn). However, this mechanism is normally based on an embedding matrix, which is not the case here for X is a simple BoW representation of the documents.
> *A1.1:* First, we would like to clarify that some parts, including the notation table, training and testing algorithms, illustration of our settings, and some derivations, are put into the supplementary materials due to the space limit. Secondly, we explain your questioning of the self-attention mechanism. We must apologize for the mistake in Eq. 5 due to our carelessness. Indeed, our self-attention operation does perform on an embedding matrix $\mathbf{H}^{(i)}\in \mathbb{R}^{D \times J}$, which is obtained by feeding BoW representations $\mathbf{X}^{(i)}\in \mathbb{R}^{V\times J}$ of the documents into a multi-layer perceptron (MLP), where $V$ is the vocabulary size, $J$ is the number of documents in each task, and $D$ is the dimension of the extracted BoW features. Finally, we apply self-attention mechanism $\mathrm{Attn}()$ to aggregate the information of $J$ document features to infer the posterior of context variable $\pmb{c}^{(i)}\in \mathbb{R}^{K}$. Hence, the correct version of Eq. (5) should be written as follows (we will fix this error in the revision):
> $$ q(\pmb{c}^{(i)}|\mathbf{X}^{(i)}) = \mathcal{N}(\pmb{\mu}\_{\pmb{c}^{(i)}}, \pmb{\Sigma}\_{\pmb{c}^{(i)}}); \pmb{\mu}\_{\pmb{c}^{(i)}}, \pmb{\Sigma}\_{\pmb{c}^{(i)}} =\mathrm{Attn}(\mathbf{H}^{(i)}); \mathbf{H}^{(i)}=\mathrm{MLP}(\mathbf{X}^{(i)}).$$
>
> - *Q1.2:* Why using the Weibull distribution in this context? There is no explanation at all.
> *A1.2:* Sorry, we do not find any part of our model involved in the Weibull distribution. But one of our compared works, Meta-SawETM [1], utilizes the Weibull distribution for posterior approximation.
>
>
> - *Q1.3:* What "latent indication" means for H(i)?
> *A1.3:* Please refer to our answer to *Q1.1*, we call the embedding matrix $\mathbf{H}^{(i)}\in \mathbb{R}^{D \times J}$ as "latent indication", which stands for the extracted deterministic features of BoW representations.
>
> - *Q1.4:* I guess the step 1.c line 114 is done after having computed all the word embedding of the generated document (we need all the Z(i)). Is it true? If so, this step should be taken outside the for loop.
> *A1.4:* Yeah. You are correct in your understanding, and step 1.c (line 114) should be taken outside the for loop. Thank you for pointing out this error, and we will fix it in the revision.
>
>
> **Q2:** Would it be possible to use the adaptive word embedding module (Z(i)) for the bag-of-word encoder?
>
> **A2:** The question you posed is a bit confusing to us. For most neural topic models, the BoW encoder is typically used to extract document representations (or topic proportions) based on the word frequencies, while the adaptive word embeddings $\mathbf{Z}^{(i)}$ in our paper are derived by modeling a task-specific semantic graph with a variational GAE. We do not understand exactly what it means to use the adaptive word embedding module for the bag-of-word encoder. Do you mean directly using the variational GAE combined with the semantic graph to learn the document representations (or topic proportions)?
>
> **Q3:** The experimental part of the paper looks quite strong. Important solutions for topic modeling (old and new) are considered ... However, I would expect more experiments on the few-shot context, for instance, by varying the number of available documents (not only 5 or 10) and taking into account the semantic difference between the corpora (i.e., between the training set and the test set).
>
> **A3:** For varying the number of available documents for each task, we conduct additional experiments on all four datasets with \{20, 50, 100\} documents in each task and list the perplexity results of different compared methods in the **Author Rebuttal by Authors** part. As for taking the semantic difference between the training set and the test set into account, we have not performed such experiments due to the time limit, but we will leave it as a priority for our future work.
>
> [1] Bayesian deep embedding topic meta-learner. In ICML 2022.

---

> > ### Comment · Reviewer_CvqN · 2023-08-11
> > **All is ok for me**
> >
> > I've read the other reviews and the rebuttal. For A1.2, you're right: I was confused and thought you used something from another paper, so it's all good to me. For A2, I better understand now that you've two types of information, so you cannot use one straight in replacement of the other, so it's ok as well. Thank you for the additional experiments which confirms the value of your work. For me it's still an "accept".

---

> > > ### Author Response · Authors · 2023-08-15
> > >
> > > We genuinely appreciate your recognition of our work! We will make the best effort to improve the presentation of our paper.
> > >
> > > Best regards.

---

### Official Review · Reviewer_VS4z · 2023-07-07

**Soundness:** 2 fair
**Presentation:** 2 fair
**Contribution:** 2 fair
**Rating:** 5
**Confidence:** 2

**Summary:**

This paper proposed an approach for few-shot topic modeling. The authors first question the limitations of "static word embeddings" in previous related work when transferring to new tasks, and then propose to use the "adaptive word embeddings" generated by VGAE to address this issue. Although the problem this paper aims to solve is meaningful, I think the description of the motivation is not clear enough. Even though the experimental results are acceptable, the methods used do not reflect the motivation of the paper.

**Strengths:**

- Few-shot learning for topic modeling is a meaningful problem.

- By combining VGAE with the neural topic model, fairly good experimental results were achieved.

**Weaknesses:**

	This paper aims to address the limitations of "static word embeddings". In line 9, the authors claim that 'we introduce a variational graph autoencoder to learn task-specific word embeddings based on the dependency graph refined from the context of each task'. However, why would the dependency graph and VGAE help in learning task-specific word embeddings? If this claim holds true, simply using the context of each task could achieve the same purpose. In other words, I don't see how VGAE and the dependency graph would aid in learning the so-called 'adaptive word embeddings'. The authors should carefully explain why the dependency graph and VGAE can reflect the characteristics of each task. Would the dependency graph for each task be fundamentally different? In summary, I find this motivation unconvincing.

	In line 14, the authors state that the Gaussian mixture prior can "facilitate the discovery of diverse topics and the quick adaptation to novel tasks." However, why using a Gaussian mixture prior can help the fast adaptation to novel tasks? What advantages does it have over the methods used in existing neural topic models? The authors should explain the principles that make the Gaussian mixture prior to work, that is, what the motivation is for using a Gaussian mixture prior, rather than vaguely stating that it can help with adaptation. I agree that the authors provide a perspective on learning topics through clustering by using a Gaussian mixture prior, but the motivation they explained is not convincing.

	Some sentences are overly long, making this paper hard to follow. For instance: Lines 24-27.

	In lines 57-66, the author's description of the contributions of this paper. The first and the second points are repetitive.

	In lines 45-46, "it is experimentally found ...". This statement is vague and confusing. If it's a finding from the experiments conducted in this paper, then the results should be presented. If it's an experimental finding from previous work, then a reference should be provided.

	According to the introduction and related work, this paper is an important existing work on few-shot topic modeling: "Few-shot learning for topic modeling". So why did the authors not compare the methods of this work in their experiments? The authors should give a reason.

	What does ϕ stand for in Equation 10? This paper does not give an explanation.

	The author demonstrated the effectiveness of the proposed method in the experiment. However, I believe there are some flaws in the experiment. Firstly, since the author believes that VGAE is the main reason for acquiring "adaptive word embeddings", then the visualization results of the model without VGAE should also be provided in Figure 3, in order to prove the role of VGAE in learning "adaptive word embeddings", which is a major motivation of this paper.

	Secondly, the ablation study is not comprehensive enough. On the one hand, the author should provide the results on all four datasets. On the other hand, the author should also provide the results of separately removing the context variable, graph vae, and GMM prior, in order to clearly prove the effectiveness of each module proposed in this work.


**Questions:**

Please see the Weaknesses.

**Limitations:**

Please see the Weaknesses.

---

> ### Author Rebuttal · Authors · 2023-08-08
>
> We sincerely appreciate your careful consideration and valuable comments. In the following, we are going to try our best to address your concerns.
>
> **W1:** This paper aims to address the limitations ... The authors should carefully explain why the dependency graph and VGAE can reflect the characteristics of each task. Would the dependency graph for each task be fundamentally different? In summary, I find this motivation unconvincing.
>
> **R1:** Above all, we would like to clarify that our method is able to learn "adaptive word embeddings" because we utilize contextual information that is closely related to each task. In our problem setup, each task consists of only a handful of documents from a specific domain, which could lead to a world of differences between the contexts of different tasks. For instance, in a task related to "hardware", the context of the word **bus** is most likely to cover the words "data", "transmitted", "cache", and so on. Whereas in a task associated with "autos", the words that most often co-occur with the word **bus** are probably "car", "taxi", "passenger", etc. An illustrated example can be found in our Supplementary Materials. We believe that this contextual information is very instrumental in capturing the precise meanings of words, so we refine it into a dependency graph and are able to learn "adaptive word embeddings" with the help of the VGAE. In other words, the dependency graph incorporating task-specific contextual information can reflect the characteristics of each task.
>
> **W2:** In line 14, the authors state that the Gaussian mixture prior can ... However, why using a Gaussian mixture prior can help the fast adaptation to novel tasks? What advantages does it have over the methods used in existing neural topic models? The authors should explain the principles ... but the motivation they explained is not convincing.
>
> **R2:** To explain clearly the motivation for using a GMM prior, we address two key questions. **1)** why do we use a variational graph autoencoder instead of a graph autoencoder? **2)** why do we adopt a Gaussian mixture prior instead of a standard normal prior? Actually, all of these choices serve one purpose, *i.e.,* to alleviate the issue of "*topic collapsing*". We found that if the learning of topic embeddings is not constrained, it will lead to highly repetitive topics (the content of all topics tends to be the same). Therefore, we use a variational GAE in the hope of regularizing the word latent space. Then we use Gaussian embeddings to represent topics to model the uncertainty, with a standard normal distribution as the prior. Even so, we found the learned topics are not diverse enough; thus, we rely on the GMM prior to further overcome the problem. **Please refer to our newly added one-page PDF for corresponding visualization and numerical results.**
>
> **W3:** Some sentences are overly long, making this paper hard to follow. For instance ... If it's an experimental finding from previous work, then a reference should be provided.
>
> **R3**: We agree with you about these flaws in writing and expression, and we will improve our presentation in the revision to make it easier to follow.
>
> **W4:** According to the introduction and related work, this paper is an important existing work on few-shot topic modeling: "Few-shot learning for topic modeling". So why did the authors not compare the methods of this work in their experiments? The authors should give a reason.
>
> **R4:** We found that the authors of "Few-Shot Learning for Topic Modeling" did not open-source their code. Indeed, we have also tried to contact the authors by email but did not get any response, so we do not compare the method of this work in our experiments. However, we have compared a more recently developed approach with strong performance – "Bayesian Deep Embedding Topic Meta-Learner".
>
> **W5:** What does $\phi$ stand for in Equation 10? This paper does not give an explanation.
>
> **R5:** We apologize for our carelessness. $\phi$ in Eq. 10 stands for the topic-word matrix, which is actually denoted by $\boldsymbol{\beta}$ in our method.
>
> **W6:** The author demonstrated the effectiveness of the proposed method in the experiment. However, I believe there are some flaws in the experiment. Firstly ... to prove the role of VGAE in learning "adaptive word embeddings", which is a major motivation of this paper.
>
> **R6:** We think there may be some misunderstanding about the role of VGAE. Indeed, the task-specific dependency graph is the main reason for acquiring "adaptive word embeddings" (refer to our response to **W1**). And VGAE is a mapping function that serves as a bridge. If we remove the VGAE module from our model, then contextual information relevant to each task will not be exploited, as the dependency graph is the input to VGAE. However, introducing additional contextual information closely related to each task is precisely the biggest innovation and contribution of our method. If we do not use the dependency graph and only rely on the BoWs of the documents to learn our model, it should perform comparably to the baseline method ETM. Since there is no information increment in this case, the only difference between our model and ETM is the consideration of one more context variable.
>
> **W7:** Secondly, the ablation study is not comprehensive enough. On the one hand, the author should provide the results ... in order to clearly prove the effectiveness of each module proposed in this work.
>
> **R7**: As you mentioned that the ablation study is not comprehensive enough, we have further perfected the ablation study, and the corresponding results are exhibited in our newly added one-page PDF. Please note that the results we report for separately removing the graph vae do not mean that we have completely discarded the VGAE module but have replaced it with a graph autoencoder. The reason for doing so has been explained in our response to **W6**.

---

> > ### Comment · Reviewer_VS4z · 2023-08-15
> >
> > Thanks for the detailed response. Most of my concerns have been resolved. I am increasing the rating to 5.

---

> > > ### Author Response · Authors · 2023-08-15
> > >
> > > Thank you for considering a higher rating. We believe the constructive feedback will help to further improve the quality of our paper.
> > >
> > > Best regards.

---

### Official Review · Reviewer_4MPJ · 2023-07-10

**Soundness:** 3 good
**Presentation:** 3 good
**Contribution:** 3 good
**Rating:** 5
**Confidence:** 5

**Summary:**

The authors target the problem of multi-meaning words across different tasks for topic models, particularly under low-resource settings. To this end, they propose a variational graph autoencoder with a trainable Gaussian mixture prior to capture the distribution of task-specific word embeddings.

**Strengths:**

Overall, the paper is sound with several strengths:

- The authors address the low-resource regime that has received little attention from the topic modeling research recently.

- The experiments are encyclopedically covered with detailed discussions.

**Weaknesses:**

However, the paper exhibits some weaknesses:

- The examples illustrating the applications of few-shot neural topic models are not persuasive. In particular, few-shot topic modeling might not be employed to learn users’ past purchases or online behaviors. In addition, during crises, e.g. Covid-19, the number of documents towards certain topics would rather burgeon than remain limited, hence invalidating the need for few-shot topic models.

- The paper needs more comparison with prior embedding-based topic models [1, 2]. Additional evaluation of the model performance against such works could make the experiments more convincing.

- The choice of prior distribution plays an important role in specifying the latent space. In the paper, the authors have not elaborated on why the Gaussian mixture model is selected as the prior, or provided experiments to demonstrate its advantage over other prior choices.

[1] Neural models for documents with metadata (Card et al., 2018)

[2] Neural topic models via optimal transport (Zhao et al., 2021)

**Questions:**

- Could you supply more performance comparison of the proposed method with the previous one, especially those utilizing word embeddings for topic modeling?

- Could you in more detail discuss the significance of the Gaussian mixture model as the prior distribution or clarify its benefits over other prior distributions?

---

> ### Author Rebuttal · Authors · 2023-08-08
>
> Thank you for your careful readings and valuable comments. We believe the constructive feedback will improve the paper and increase its potential impact on the community. Regarding the weaknesses you mentioned, we respond as follows.
>
> **W1:** The examples illustrating the applications of few-shot neural topic models are not persuasive. In particular, few-shot topic modeling might not be employed to learn users’ past purchases or online behaviors. In addition, during crises, e.g. Covid-19, the number of documents towards certain topics would rather burgeon than remain limited, hence invalidating the need for few-shot topic models.
>
> **R1:** We are sorry that you think the examples illustrating the applications of few-shot neural topic models are not persuasive enough. But the few-shot topic models do have their own value in real-world applications. Take the example you listed. To this day, there has indeed been a substantial increase in the number of documents on the topic of "Covid-19". However, at the very beginning of this epidemic, there were relatively few cases of infection in all regions, and the reports of related medical diagnoses were very limited. Under such circumstances, few-shot topic models can help us extract key information from limited resources, thus facilitating us to take appropriate preventive and control measures.
>
> **W2:** The paper needs more comparison with prior embedding-based topic models [1, 2]. Additional evaluation of the model performance against such works could make the experiments more convincing.
>
> **R2:**  Based on your suggestion, we compared our model with SCHOLAR [1] and NSTM [2], two prior embedding-based topic models on all four datasets with document size 10. The results are listed in the following.
>
> |20NG|PPL|TD|TC|
> |:--|:--:|:--:|:--:|
> |ETM [a]|3107|0.8395|-0.8437|
> |Meta-SawETM [b]|2984|0.5643|-0.6086|
> |SCHOLAR [1]|4371|0.5779|-0.6792|
> |NSTM [2]|3190|0.7008|-0.6202|
> |Meta-CETM|1170|0.8154|-0.3701|
>
> |Yahoo|PPL|TD|TC|
> |:--|:--:|:--:|:--:|
> |ETM [a]|2817|0.8851|-0.8913|
> |Meta-SawETM [b]|2365|0.5465|-0.6406|
> |SCHOLAR [1]|4697|0.4981|-0.7429|
> |NSTM [2]|3153|0.7505|-0.6351|
> |Meta-CETM|1219|0.7886|-0.4639|
>
> |DB14|PPL|TD|TC|
> |:--|:--:|:--:|:--:|
> |ETM [a]|3054|0.8106|-0.8719|
> |Meta-SawETM [b]|1914|0.7545|-0.9204|
> |SCHOLAR [1]|4913|0.5112|-0.8217|
> |NSTM [2]|3379|0.6195|-0.7626|
> |Meta-CETM|1084|0.7475|-0.4783|
>
> |WOS|PPL|TD|TC|
> |:--|:--:|:--:|:--:|
> |ETM [a]|3310|0.9286|-0.9785|
> |Meta-SawETM [b]|2253|0.7217|-0.7475|
> |SCHOLAR [1]|3884|0.6413|-0.7434|
> |NSTM [2]|3164|0.7659|--0.6472|
> |Meta-CETM|1293|0.8667|-0.5177|
>
> **W3:** The choice of prior distribution plays an important role in specifying the latent space. In the paper, the authors have not elaborated on why the Gaussian mixture model is selected as the prior, or provided experiments to demonstrate its advantage over other prior choices.
>
> **R3:** To explain clearly the motivation for using a GMM prior, we address two key questions. **1)** why do we use a variational graph autoencoder instead of a graph autoencoder? **2)** why do we adopt a Gaussian mixture prior instead of a standard normal prior? Actually, all of these choices serve one purpose, *i.e.,* to alleviate the issue of "*topic collapsing*". We found that if the learning of topic embeddings is not constrained, it will lead to highly repetitive topics (the content of all topics tends to be the same). Therefore, we use a variational GAE in the hope of regularizing the word latent space. Then we use Gaussian embeddings to represent topics to model the uncertainty, with a standard normal distribution as the prior. Even so, we found the learned topics are not diverse enough; thus, we rely on the GMM prior to further overcome the problem. **Please refer to our newly added one-page PDF for corresponding visualization and numerical results.**
>
>
> [1] Neural models for documents with metadata (Card et al., 2018)
>
> [2] Neural topic models via optimal transport (Zhao et al., 2021)
>
> [a] Topic modeling in embedding spaces (Dieng et al., 2020)
>
> [b] Bayesian deep embedding topic meta-learner (Duan et al., 2022)

---

> > ### Comment · Reviewer_4MPJ · 2023-08-19
> > **Thank you for the rebuttal**
> >
> > Thank you the authors for their response. After reading all reviews and rebuttals I found that some of my concerns have been resolved to some extent. However, because the examples are still equivocal to me, I decide to keep my scoring.

---

> > > ### Author Response · Authors · 2023-08-21
> > >
> > > Thank you for your further reply. As you mentioned that some of your concerns have been resolved, we would like to know which ones you still have that were not adequately addressed. Also, please let us know which examples are still equivocal to you. Do you mean the examples used to illustrate the applications of few-shot neural topic models?

---

### Official Review · Reviewer_biyw · 2023-07-10

**Soundness:** 3 good
**Presentation:** 2 fair
**Contribution:** 3 good
**Rating:** 5
**Confidence:** 2

**Summary:**

The authors present a new neural topic model which aims to solve the problem of learning task-specific word embeddings in a low resource scenario.  In addition to a somewhat typical neural TM, dependency graphs are collected using parsers, and embedded via GCNs to produce adaptive word embeddings.  A topic-word matrix then models the task-specific distribution over words as a gaussian over their adaptive embeddings.  The topic-specific embedding spaces capture the role a word is likely to play within that context.  The model is evaluated across 4 datasets and a number of metrics, where it shows strong improvements in perplexity, and good performance in document classification.

**Strengths:**

- strong empirical performance in perplexity, and good performance in document classification.

- comparisons to many existing similar models.  The improvements over models using sentence BERT are especially interesting.  Good ablations.

**Weaknesses:**

- reliance on pre-trained parsing tools.  How well does this approach work in different languages, or styles/domains of text that differ significantly from the parsing training data?

- the problem at the heart of this work is lexical ambiguity.  Here parsing, and then graph-based embedding of the parse, aim to find task-specific meanings of each word, and there is little doubt that adaptive word embeddings are an effective approach to this problem.  Outside of topic modeling, LLMs and other transformer-based models solve this same task, and also refine word-specific embeddings into context-specific ones.  The only aspect in which this is included is via CombinedTM/ZeroShotTM.  Admittedly, I haven't worked on topic modeling since the switch to neural models, but it would seem that BERT would also solve the issues being pursued in this work.  I didn't see a compelling explanation for why this approach based on dependency graphs is an improvement over generic transformer-based sentence embeddings, all other aspects remaining the same.

- even if the work is geared towards low resource settings, I would like to see the performance as a factor of task dataset size.  In its current presentation I think the performance is compelling enough to warrant acceptance, as the performance improvement seems sufficient across a number of metrics to be of great interest to the topic modeling community.  However, if those performance margins decrease significantly (or completely) as dataset size increases to even modest sizes, it would be difficult to find a use-case for this approach.  It would be the story of perhaps offloading some of the learning problem to the pre-trained parsing and priors over their embeddings, which solves a practical problem, but not so much one of academic interest.

- discussion of conceptually related work is sparse.  The related works section is overly brief given the amount of work that solving similar core tasks.  One earlier work that seems spiritually related might be [1].

- surprisingly CNN > MLP is a more important design decision than any architectural change represented across several previous papers.  It raises a little bit of an alarm bell regarding the importance of these architectural designs as evaluated here.  Extending the document classification task to more datasets (than the 2 here) would increase confidence in the importance of the work.

L26: vontextualized

[1] Word Representations via Gaussian Embedding
https://arxiv.org/abs/1412.6623



**Questions:**

(mixed with weaknesses)

---

> ### Author Rebuttal · Authors · 2023-08-09
>
> We sincerely appreciate your careful consideration and valuable comments. In the following, we are going to try our best to address your concerns.
>
> **W1:** reliance on pre-trained parsing tools. How well does this approach work in different languages, or styles/domains of text that differ significantly from the parsing training data?
>
> **R1:** Thank you very much for raising a very meaningful question. However, we think it is outside the scope of this paper, and we will keep it as a focus of our future work.
>
> **W2:** The problem at the heart of this work is lexical ambiguity ... I didn't see a compelling explanation for why this approach based on dependency graphs is an improvement over generic transformer-based sentence embeddings, all other aspects remaining the same.
>
> **R2:** On the one hand, the remarkable capabilities of pre-trained language models such as BERT rely on the massive training corpus (e.g., BERT was trained on a dataset of over 3.3 billion words). In contrast, we present a data-efficient framework that only relies on a batch of training tasks (about 5 million words for 20NG) to produce well-fitted word embeddings for each task. On the other hand, due to the strong biases introduced by the training corpus (even though it may cover a wide range of content), pre-trained language models may not be so effective in adapting to a completely unfamiliar context. Whereas our proposed framework has been carefully designed to learn how to effectively adapt to novel tasks from an unfamiliar corpus, it should perform better as a few-shot learner.
>
> **W3:** Even if the work is geared towards low resource settings, I would like to see the performance as a factor of task dataset size ... However, if those performance margins decrease significantly (or completely) as dataset size increases to even modest sizes, it would be difficult to find a use-case for this approach. It would be the story of perhaps offloading some of the learning problems to the pre-trained parsing and priors over their embeddings, which solves a practical problem, but not so much one of academic interest.
>
> **R3:** We have conducted additional experiments by varying the number of documents in each task from 20 to 50 to 100. Please refer to our unified response to all reviewers (**Author Rebuttal by Authors**) to see the corresponding results.
>
> **W4:** Discussion of conceptually related work is sparse. The related works section is overly brief given the amount of work that solving similar core tasks. One earlier work that seems spiritually related might be [1].
>
> **R4:** We agree that there has been sparse discussion of conceptually related work, and we will add more abundant discussions about the corresponding works in the revision, including the one [1] you mentioned.
>
> **W5:** surprisingly CNN > MLP is a more important design decision than any architectural change represented across several previous papers. It raises a little bit of an alarm bell regarding the importance of these architectural designs as evaluated here. Extending the document classification task to more datasets (than the 2 here) would increase confidence in the importance of the work.
>
> **R5:** Yeah, we are also surprised by the experimental results showing that CNN > MLP is a more important design decision in our setup, perhaps because our input data is the bag-of-words (BoWs) representations of the documents. Anyway, to increase the credibility of this finding, we conduct document classification experiments on additional datasets, *i.e.,* Yahoo and WOS. The results are listed in the following table.
>
> ||Yahoo|||WOS||
> |:--|:--:|:--:|:--:|:--:|:--:|
> |**Methods**|**5way-5shot**|**5way-10shot**||**5way-5shot**|**5way-10shot**
> |MAML (MLP)|45.42|51.00||37.77|40.43|
> |PROTO (MLP)|50.01|56.16||39.61|41.46|
> |FT (MLP)|48.59|53.06||36.52|37.22|
> |FT$^{*}$ (MLP)|50.73|56.74||45.02|51.20|
> |MAML (CNN)|48.81|56.50||47.28|57.32|
> |PROTO (CNN)|53.16|63.66||59.05|**67.75**|
> |FT (CNN)|56.78|66.04||54.68|63.39|
> |FT$^{*}$ (CNN)|53.28|52.56||51.42|61.98|
> |HNS-SawETM|52.35|57.86||42.09|56.91|
> |Meta-SawETM|52.45|60.58||43.39|57.44|
> |CombinedTM|57.94|64.75||56.16|65.97|
> |ZeroShotTM|58.12|66.21||58.50|66.10|
> |Meta-CETM|**63.84**|**72.67**||**61.47**|67.62|
>
> [1] Word Representations via Gaussian Embedding https://arxiv.org/abs/1412.6623

---

> > ### Author Response · Authors · 2023-08-21
> >
> > Dear **Reviewer biyw**,
> >
> > Thanks for your patience and careful review! We have endeavored to address your concerns at the first rebuttal stage, and hope that those responses and empirical results will further convince you of the significance of our work.
> >
> > Considering that the discussion period will end on **Aug 21st**, we would like to know if you have any other questions about our paper, and we are glad to have a discussion with you during the remaining time.
> >
> > Best regards.

---

> > > ### Comment · Reviewer_biyw · 2023-08-21
> > > **Thank you for you rebuttal**
> > >
> > > Thank you authors for the rebuttal with additional experiments on W5.
> > >
> > > The results look good and I think it does strengthen the paper overall.  I do feel that the W2(+ architectural aspects of W5) creates some uncertainty over the staying power of this approach, but since they are difficult questions to address without writing an entirely different paper, and because I don't think that necessarily undermines the solid contributions of the work presented in the current paper, I am inclined to keep my current recommendation on the side of accept.
> > >
> > > In my opinion, a necessary aspect of raising the score further would be exploration of the concerns in W1, which would show a real practical usefulness of this approach outside of more academically-oriented datasets which are limited in terms of genre and style.  I respect not wanting to present the paper as is, but this is the age old issue with parser-based approaches, namely, how much do they break when applied to datasets which do not look like the parsing training data (which is naturally often more limited than raw text resources).  I think it is also reasonable to expect it to be addressed in this paper, though my inclination says that the way in which you use the parse trees does not seem to rely on precise syntactic understanding and may be robust to errors arising from the parser.

---

> > > > ### Author Response · Authors · 2023-08-21
> > > > **Thanks for your constructive suggestion**
> > > >
> > > > We appreciate your insight into the robustness of our approach. We also understand your concern that parser-based methods might break when applied to datasets that do not look like the parsing training data. However, we would like to emphasize that **the most important reason that makes our approach effective is the introduction of task-relevant contextual information, which helps to capture the precise meanings of words in each task.** Although we currently leverage dependency parsing tools to represent the contextual information as a semantic graph, the way of constructing semantic graphs can actually be flexible, *e.g.,* sliding windows. Consider the following context, "*...there are so many privately-owned cars, as well as taxis and buses, the rush-hour has become a 24-hour phenomenon...*", whether we resort to dependency parsing or sliding window, the words "car", "taxi" and "bus" will be connected in the semantic graph, which indicates that the meaning of "bus" here refers to a vehicle rather than an electrical conductor. Therefore, we believe that our approach does not rely on precise syntactic understanding and may be robust to errors arising from the parser. Due to the time limit, we do not have the relevant experimental results for the time being, but we promise to conduct subsequent experiments to further confirm the robustness of our method, which will be presented in the appendix of the final version.

---

> > > > > ### Comment · Reviewer_biyw · 2023-08-21
> > > > > **Regarding the example**
> > > > >
> > > > > Thank you for the example, but I think this is precisely what I was referring to in my earlier review in wondering what the dependency graphs are bringing to this work that is lacking in various LLM representations.  In the example you give, of course we expect BERT or other similar transformer-based architectures to capture exactly the same thing, without any reliance on another tool, with the natural robustness to a wide variety of genres that comes from being trained on huge amounts of diverse in-the-wild text.  I believe the idea that many existing LLMs already solve this ambiguity issue, and quite well, is somewhat of a common knowledge but I'm all ears to hearing any evidence to the contrary.  This is why I ultimately feel that the proposed approach will not be necessary, even at this point in time, if this is truly where the power of the proposed method lies.

---

> > > > > > ### Author Response · Authors · 2023-08-21
> > > > > > **Regarding the LLMs**
> > > > > >
> > > > > > Thanks for your advice. Exactly what you said, the natural robustness of various LLM representations (including BERT or other similar transformer-based architectures) comes from being trained on huge amounts of diverse in-the-wild text. And what we want to explore is an alternative when there is no access to sufficient and diverse training data. Under such circumstances, how we can still achieve robustness and generalization capabilities that are analogous to LLMs is precisely the significance of our work. Thus, the inner workings of our framework do not lie in training with huge amounts of in-the-wild text, but in learning to adapt to new tasks from unfamiliar corpora effectively using limited training resources. We understand the popularity of LLMs since the birth of ChatGPT, they appear to be capable of seemingly anything. However, we call for a broader range of possibilities to be explored beyond this, which would be better for the community's long-term development.

---

### Official Review · Reviewer_fyGx · 2023-07-23

**Soundness:** 4 excellent
**Presentation:** 3 good
**Contribution:** 3 good
**Rating:** 7
**Confidence:** 3

**Summary:**

This paper addresses the problem of inducing neural topic models in low-resource regimes by learning adaptive word embeddings that exploit contextual grammar information. The adaptive word embeddings are learnt with a variational graph autoencoder and the topics are formed from a gaussian mixture prior of the latent word embedding space that generates the observed words and their semantic relations.  The paper introduces a variational inference algorithm to estimate the document- and topic-specific latent variables using MLP and MLP with attention, respectively. In addition, GCN is employed to learn the latent word representations given initial word embeddings and word relations derived from a neural dependency parser. Finally, the parameters of the GMM are learned independently via EM. Evaluation shows strong empirical results in few-shot topic modeling and document classification.

**Strengths:**

The main contribution of the paper is the introduction of contextual semantic information into the neural topic model. Empirical results suggest that this approach is beneficial both in terms of per-holdout-word perplexity and topic coherence while remaining competitive in terms of topic diversity. Strong results are also observed in few-shot classification. The paper is fairly well-written but the baselines could have been better explained either in section 3.1 or the related work.


**Weaknesses:**

I cant parse eq. 10 in the evalution: what is the superscript (1)s?
In the experiments in section 3.2.3 (few-shot classification), I could not follow the evaluation scheme described in lines 238-239. How were these experiments conducted? How were the class-specific topic-word matrices calculated, what data was used for this, and how was the reconstruction error measured? Also, I could not find the results from this analysis. The results in Table 2 correspond to datasets that have ground truth topic labels.

**Questions:**

See weaknesses section.

**Limitations:**

No limitations were discussed

---

> ### Author Rebuttal · Authors · 2023-08-07
>
> Thank you for acknowledging the quality of our work! We will take your suggestions to give a more detailed explanation of the baselines in the revision. Here we clarify some of your questions.
>
> **Q1:** I can't parse eq. 10 in the evaluation: what is the superscript (1)s?
>
> **A1:** Eq. 10 is the formula for calculating the per-holdout-word perplexity, where $\phi \in \mathbb{R}^{V \times K}$ denotes the topic-word matrix and $\theta \in \mathbb{R}^{K \times N}$ represents the topic proportion matrix, and the superscript $s$ indicates the index of the $i^{th}$ collected sample. Here we use the notation $\phi^{(1)s}\theta^{(1)s}$ for the consideration of hierarchical topic models, which have multiple layers of document representations. By using the superscript (1), we mean that we use the representation of the bottom layer to compute the perplexity. Indeed, for most regular topic models with only a single layer of document representation, we can omit the superscript (1) and write the notation as $\phi^{s}\theta^{s}$.
>
> **Q2:** In the experiments in section 3.2.3 (few-shot classification), I could not follow the evaluation scheme described in lines 238-239. How were these experiments conducted? How were the class-specific topic-word matrices calculated, what data was used for this, and how was the reconstruction error measured? Also, I could not find the results from this analysis. The results in Table 2 correspond to datasets that have ground truth topic labels.
>
> **A2:** Above all, the few-shot classification experiment requires available ground truth topic labels, which are used to compute the accuracy. Next, we elaborate on the evaluation scheme described in lines 238-239 with an example of a 5-way 5-shot classification task. Specifically, the data of a 5-way 5-shot task consists of a "*support set*", which includes 25 documents from 5 different topics (5 for each topic), and a "*query set*", which contains 15 documents from each of the 5 topics. Our goal is to train a task-specific classifier using the "*support set*" and compute the classification accuracy on the "*query set*". So for the well-trained topic models, we use five documents of each topic in the "*support set*" to adapt a topic-word matrix, respectively; the resulting five topic-word matrices, denoted as {$\phi_1$, $\phi_2$, $\phi_3$, $\phi_4$, $\phi_5$}, are called class-specific topic-word matrices (here a topic refers to a category). Then for each document in the "*query set*", we use the trained topic model to derive its topic proportion $\theta_q$, which is subsequently combined with each of the five class-specific topic-word matrices to calculate the data likelihood $\\{ p(x_q | \theta_q, \phi_i) \\}_{i=1}^5$, $x_q$ is the BoW of the query document. The reconstruction error is defined as the negative data likelihood, so we classify it as the topic with the smallest reconstruction error.

---

> > ### Author Response · Authors · 2023-08-21
> >
> > Dear **Reviewer fyGx**,
> >
> > Thanks for your patience and careful review! Not sure if the responses we offered in the first rebuttal stage adequately addressed your concerns?
> >
> > Considering that the discussion period will end on **Aug 21st**, please let us know if you have any other questions about our paper, and we will be happy to discuss them with you during the remaining time.
> >
> > Best regards.

---

> > > ### Comment · Reviewer_fyGx · 2023-08-21
> > >
> > > Thank you for the clarifications. I will keep my ratings.

---

> > > > ### Author Response · Authors · 2023-08-21
> > > >
> > > > We genuinely thank for your recognition of our work! We will improve the presentation of our paper in the revision.
> > > >
> > > > Best regards.

---

### Official Review · Reviewer_Jyas · 2023-07-27

**Soundness:** 3 good
**Presentation:** 2 fair
**Contribution:** 3 good
**Rating:** 5
**Confidence:** 3

**Summary:**

This paper proposes a method for few-shot topic modeling. Specifically, rather than following traditional wisdoms to learn static word embeddings for all the tasks/domains, the authors allow task-specific word representations such that the knowledge from the source task can be better transferred to a target task. The authors also employ Gaussian mixture prior with EM algorithm to capture the clustering structure of distributed word representations. Experimental results demonstrate the effectiveness of the proposed method.

**Strengths:**

1. The proposed method for few-shot topic modeling looks novel and intuitive to me.
2. The experiment looks inclusive and the proposed method achieved leading performance.

**Weaknesses:**

1. Though the paper is generally well-written, some parts are confusing to me. Please refer to the questions part.
2. [1] also follows a clustering perspective for topic modeling and the difference between [1] and the proposed method can be discussed.


[1] Effective Neural Topic Modeling with Embedding Clustering Regularization

**Questions:**

1. In Eq 1, is it $Z^{(i)\intercal}Z^{(i)}$ or $Z^{(i)}Z^{(i)\intercal}$? What is the relationship between $\hat{A}$ and $A$?
2. In the introduction line 49, the authors claim that "task-specific semantic graphs between words using well-established dependency parsing tools". However, I fail to find any information about the semantic graph using parsing tools in the rest of the paper.

**Limitations:**

The limitations are not discussed in the paper.

---

> ### Author Rebuttal · Authors · 2023-08-06
>
> We appreciate your constructive comments and feedback. The weaknesses have been addressed below.
>
> **Q1:** In Eq 1, is it  ${Z^{(i)}}^{\top}Z^{(i)}$ or $Z^{(i)}{Z^{(i)}}^{\top}$? What is the relationship between $\hat{A}$ and $A$?
>
> **A1:** Since we assume $Z^{(i)} \in \mathbb{R}^{D \times V}$ in the article, it should be ${Z^{(i)}}^{\top}Z^{(i)}$ in Eq. 1, which corresponds to the generated adjacency matrix $A^{(i)} \in \mathbb{R}^{V \times V}$. Thank you for pointing out our typo here. In addition, we use a variational graph autoencoder to model the adjacency matrix $A$ of the semantic graph, following the standard encoder-decoder architecture shown in Fig. 2, $\hat{A}$ can be viewed as the reconstruction of $A$, *i.e.*, $A$ is the ground-truth adjacency matrix and $\hat{A}$ represents the predicted adjacency matrix.
>
> **Q2:** In the introduction line 49, the authors claim that "task-specific semantic graphs between words using well-established dependency parsing tools". However, I fail to find any information about the semantic graph using parsing tools in the rest of the paper.
>
> **A2:** We apologize for not covering this part of the information, and we will add the corresponding details in the revision. Actually, we build task-specific semantic graphs with the help of **spaCy**, a library for advanced natural language processing that provides a variety of linguistic annotations to give us insights into texts' grammatical structure. Concretely, for a specific task, we use the built-in syntactic dependency parser to analyze each document, and the resulting dependency labels describe the relations between individual tokens, like a subject or object, which also become the basis for constructing the task-specific semantic graph. For example, if a dependency label is assigned between two vocabulary terms in any document, we add an edge between the corresponding nodes in the semantic graph. Conversely, if two vocabulary terms are not assigned a dependency label in all documents, then there are no edges between the corresponding nodes in the graph. Fig. 1 of our Supplementary Materials illustrates the constructed task-specific semantic graphs.
>
> **W2:** [1] also follows a clustering perspective for topic modeling and the difference between [1] and the proposed method can be discussed.
>
> **Discussion:** While both [1] and our method follow a clustering perspective to learn topics, the focus and target problems they aimed at are different. On the one hand, the starting point of [1] is the phenomenon of "*topic collapsing*", a common issue that plagues most existing topic models. And its solution is to regularize topic embeddings as cluster centers and word embeddings as cluster samples on the ground of ETM [2]. It assigns all words properly to each topic by solving a well-defined optimal transportation problem. On the other hand, our method strives to solve the problem of learning topics effectively from only a few documents. And the starting point is to learn adaptive word embeddings whose semantics can be well adapted to the given task by using extra contextual grammar information. Therefore, we posit a graph autoencoder to model the semantic dependency graph. To avoid learning repetitive topics and ensure learned topic distributions cover as many significant words as possible for the given task, we impose a GMM prior on the word latent space such that the adaptive word embeddings can be reasonably encapsulated by the topic distributions.
>
> [1] Effective Neural Topic Modeling with Embedding Clustering Regularization
> [2] Topic Modeling in Embedding Spaces

---

> > ### Comment · Reviewer_Jyas · 2023-08-14
> >
> > Thank the author for the response. After reading all the review comments and rebuttals I decide to keep my rating unchanged.

---

> > > ### Author Response · Authors · 2023-08-15
> > >
> > > Thanks for your time and valuable comments.
> > >
> > > Best regards.

---

### Author Rebuttal · Authors · 2023-08-09

We really appreciate all the reviewers for their constructive and helpful comments. And we apologize for typos, grammar mistakes, unclear notations and missing citations. They will be corrected such that the overall writing meet NeurIPS standards. Here we briefly introduce our newly added rebuttal PDF and also provide additional experimental results.

------
1. In our one-page rebuttal PDF, we exhibit the results of:

- visualizations of adapted embedding space with different prior choices for topic-word matrix $\boldsymbol{\beta}$, including **no prior**, **standard Gaussian distribution prior** and **GMM prior**, as suggested by reviewer **4MPJ** and reviewer **VS4z**.

- ablation study on all four datasets (20NG, DB14, Yahoo and WOS) of separately removing our module designs (context variable, Graph VAE and GMM prior) in Table.1, as suggested by reviewer **VS4z**. The ETM is chosen as our baseline. For removing Graph VAE but keeping the GMM prior, we replace the **Gragh VAE** with a **Graph auto-encoder** to model the task-specific dependency graph. For removing GMM prior but maintaining the Graph VAE module, we apply the **standard Gaussian distribution prior** to replace the **GMM prior**.

------
2. Moreover, as mentioned by reviewer **biyw** and reviewer **CvqN**, we vary the number of documents in each task from \{5, 10\} to \{20, 50, 100\} and provide the perplexity (PPL) results on all four datasets as the following.

|||20NG||||DB14||
|:--|:--:|:--:|:--:|:--:|:--:|:--:|:--:|
|**Methods**|**20**|**50**|**100**||**20**|**50**|**100**|
|LDA|2979|2443|2118||3095|2353|1858|
|PFA|2439|2271|**2060**||1903|1887|1637|
|ProdLDA|4807|4489|4466||5819|5794|6016|
|ETM|3276|3215|3199||2870|2834|2837|
|MAML-ProdLDA$^{*}$|4378|4372|4359||4612|4463|4381|
|MAML-ETM$^{*}$|3287|3186|3172||2819|2778|2715|
|Meta-SawETM|2657|3761|3661||2355|2577|2984|
|CombinedTM|2331|2267|2205||1863|1765|1695|
|ZeroShotTM|2673|2397|2330||1722|1638|1604|
|Meta-CETM(ours)|**1216**|**1517**|2138| |**1109**|**1306**|**1468**|

|||Yahoo||||WOS||
|:--|:--:|:--:|:--:|:--:|:--:|:--:|:--:|
|**Methods**|**20**|**50**|**100**||**20**|**50**|**100**|
|LDA|3916|3279|2833||2370|2091|1896|
|PFA|2545|2326|2169||1675|1663|**1643**|
|ProdLDA|6093|5784|5736||4617|4386|4369|
|ETM|2781|2801|2811||3189|3176|3296|
|MAML-ProdLDA$^{*}$|4033|3951|4202||3908|3863|3845|
|MAML-ETM$^{*}$|3439|3315|3256||4189|4062|3947|
|Meta-SawETM|2859|3037|3251||3620|3365|3101
|CombinedTM|2543|2481|2286||2587|2473|2330|
|ZeroShotTM|2664|2496|2319||2660|2497|2372|
|Meta-CETM(ours)|**1369**|**1440**|**1743**||**1482**|**1576**|1786|

---

### Decision · Program_Chairs · 2023-09-21

**Decision:**

Accept (poster)

**Comment:**

The paper proposes a new method for context-guided embedding adaptation for topic modeling in low-resource scenarios. All reviewers are consistent in accepting the paper, so I also recommend acceptance. However, the authors should address the reviewers' concerns in the revision.